# TCP-Diffusion: A Multi-modal Diffusion Model for Global Tropical Cyclone Precipitation Forecasting with Change Awareness

**Cheng Huang** [1]  **Pan Mu** [1]  **Cong Bai** [1 2]  **Peter A. G. Watson** [3]

## Abstract

Deep learning methods have made significant progress in regular rainfall forecasting, yet the more hazardous tropical cyclone (TC) rainfall has not received the same attention. While regular rainfall models can offer valuable insights for designing TC rainfall forecasting models, most existing methods suffer from cumulative errors and lack physical consistency. Additionally, these methods overlook the importance of meteorological factors in TC rainfall and their integration with the numerical weather prediction (NWP) model. To address these issues, we propose Tropical Cyclone Precipitation Diffusion (TCP-Diffusion), a multi-modal model for forecasting TC precipitation given an existing TC in any location globally. It forecasts rainfall around the TC center for the next 12 hours at 3 hourly resolutions based on past rainfall observations and multi-modal environmental variables. Adjacent residual prediction (ARP) changes the training target from the absolute rainfall value to the rainfall trend and gives our model the capability of rainfall change awareness, reducing cumulative errors and ensuring physical consistency. Considering the influence of TC-related meteorological factors and the useful information from NWP model forecasts, we propose a multi-model framework with specialized encoders to extract richer information from environmental variables and results provided by NWP models. The results of extensive experiments show that our method outperforms other DL methods and the NWP method from the European Centre for Medium-Range Weather Forecasts (ECMWF).

[1]College of Computer Science, Zhejiang University of Technology, Hangzhou, China [2]Zhejiang Key Laboratory of Visual Information Intelligent Processing, Hangzhou, China [3]School of Geographical Sciences, University of Bristol, Bristol, United Kingdom. Correspondence to: Cong Bai <congbai@zjut.edu.cn>.

*Proceedings of the $42^{nd}$ International Conference on Machine Learning*, Vancouver, Canada. PMLR 267, 2025. Copyright 2025 by the author(s).

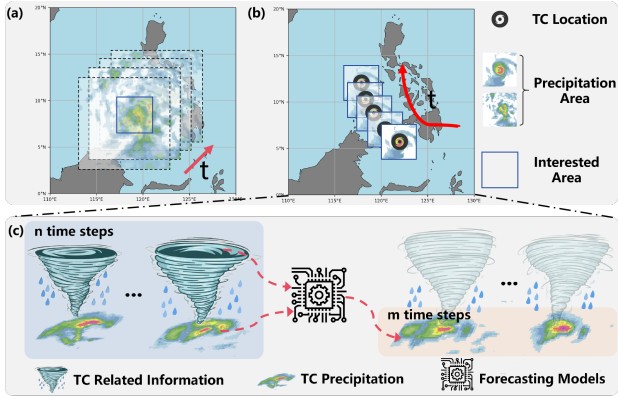

*Figure 1.* The difference between the regular precipitation forecasting (a) and TC (b) precipitation forecasting. Regular precipitation forecasting (a) usually focuses on a specific fixed region, where the predicted area (blue box) does not move with the rainfall area (black dashed boxes). In TC precipitation forecasting (b), the primary region of interest (blue boxes) moves with the TC. Sub-figure (c) represents the main prediction process of our model. The light blue box represents historical data, which is considered to be the input of the forecasting model. The light orange box represents the TC rainfall that should be predicted. This paper just focuses on the prediction of TC rainfall.

## 1. Introduction

Tropical cyclone (TC) precipitation refers to substantial rainfall events that accompany the formation, development, and movement of TCs. These precipitation events can lead to severe disasters in the affected regions, like flooding, mudslides, and landslides. These TC rainfall-related disasters cause more economic losses and deaths than strong winds (Wendler-Bosco & Nicholson, 2022), yet most TC forecasting research focuses only on TC track and intensity while neglecting the TC precipitation. If the TC precipitation can be accurately predicted, preemptive measures may be taken to mitigate the disasters brought by TC rainfall, protecting people's lives and property. Therefore, it is important to accurately predict the precipitation around the TC center.

Numerical weather prediction (NWP) is widely used for various weather forecasting tasks, including TC rainfall

forecasting. It simulates atmospheric dynamics, thermodynamics, and physical processes by solving the physical and mathematical equations that describe changes in wind, temperature, humidity, and pressure (Bai et al., 2020). It has achieved impressive results, but the development in NWP research has been gradual (Benjamin et al., 2018). Additionally, not all physical mechanisms of TCs are fully understood, which prevents the formulation of perfect physical and mathematical equations for NWP (Xu et al., 2024). Furthermore, NWP typically runs on supercomputer platforms, consuming extensive computational resources and time for predictions, when predicting a high-resolution result. Nonlinear simulation also poses a significant challenge for NWP. Fortunately, the development of deep learning (DL) offers solutions to these NWP challenges. DL-based methods excel at modeling nonlinear processes, and the cost of model training and inference is relatively low, requiring only a few Graphics Processing Units for regular AI tasks.

DL methods have been applied to many meteorological tasks, like TC track and intensity prediction (Huang et al., 2022; 2023; 2025; Zhang et al., 2025), TC rainfall downscaling (Vosper et al., 2023), global meteorological variable prediction (Bi et al., 2023; Lam et al., 2023), and precipitation nowcasting (Bai et al., 2022; Yang et al., 2024). For regular precipitation nowcasting, several works have used DL. Some previous works used a U-Net architecture network to predict rainfall (Gao et al., 2022; Xu et al., 2021). This could perform well on metrics like mean squared error, but gave predictions that were spatially smoother than real TC rainfall. Some research used the generative adversarial network (GAN) model to obtain more realistic rainfall predictions (Tian et al., 2019). Recently, due to issues with mode collapse and training instabilities in GAN models (Gao et al., 2024), researchers have applied diffusion models (DMs) for better rainfall prediction (Asperti et al., 2023; Addison et al., 2022; 2024).

These previous precipitation forecast models provide a good reference for constructing TC precipitation models. However, there are notable differences between the TC precipitation forecasting task that we consider here and regular precipitation forecasting, as shown in Figure 1.(a) and (b). For example, regular precipitation forecasting usually focuses on a specific fixed region, where the area being predicted does not move with the rainfall. In contrast, in this work, we focus on forecasting the TC precipitation, where the primary region of interest and the corresponding environmental factors, such as terrain, will change dynamically with the movement of the TC. Therefore, it is valuable to investigate whether better methods can be developed for this task. We consider the three following approaches to improving TC rainfall prediction skills:

**Changing the training goal:** most precipitation forecast-

ing methods predict the absolute value of rainfall. However, future rainfall can be understood as the sum of the current rainfall and change in rainfall ($\Delta Rainfall_{Future}$) over time, which is shown in Figure 2.(c). Specifically:

$$Rainfall_{Future} = Rainfall_{Current} + \Delta Rainfall_{Future} \tag{1}$$

We can train a DL model to predict $\Delta$ Rainfall and use Equation 1 to obtain the absolute value of future rainfall, rather than predicting the absolute value of future rainfall directly. This mechanism is advantageous because predicting the change in TC rainfall not only helps reduce cumulative errors but also ensures physical consistency–meaning that changes in rainfall intensity and spatial patterns should align to some extent with historically observed trends. We define this capability of a model as change awareness. Similar mechanisms are used in some NWP methods (Kalnay, 2003) and global weather forecasting models (Price et al., 2025; Nguyen et al., 2024; Oskarsson et al., 2024; Hu et al., 2023) to improve the accuracy and stability of meteorological forecasts. Therefore, it is worth considering this mechanism in the TC rainfall prediction task.

**Extracting richer meteorological information:** most previous DL precipitation forecasting methods attempt to extract sufficient information solely from the rainfall value data. However, the information in rainfall data alone is not adequate for DL models to learn the patterns of TC rainfall. Various TC-related data can help people and DL models understand and predict the tendency of TC rainfall, shown in the light blue box in Figure 1.(c). For instance, since rainfall is wind-driven, TC wind distribution is critical. Thus, beyond direct rainfall data extraction, modeling synergistic effects of meteorological factors (e.g., wind-rainfall dynamics) is also essential to capture TC rainfall mechanisms.

**Integrating with NWP models:** DL models can make use of global NWP predictions to enhance forecasts of specific phenomena (Harris et al., 2022; Zhu et al., 2022; Antonio et al., 2024). If the NWP forecasts are produced using physical equations, such postprocessing approaches allow useful information to be extracted from the embedded understanding. Therefore, we evaluate using specific information from NWP results to guide better predictions.

Based on these above potentially useful ideas, we propose TCP-Diffusion, a multi-modal diffusion model for global tropical cyclone precipitation forecasting with change awareness. This model can simultaneously forecast TC rainfall for multiple future time steps, at 3-hourly intervals, within the next 12 hours. The contributions are summarized as follows:

- TCP-Diffusion can extract information from a rich

set of meteorological variables, learn the rules of TC rainfall development, and make skilful predictions for global tropical cyclones. To our knowledge, it is the first global TC precipitation forecasting work based on DL.

- Adjacent Residual Prediction (ARP) is proposed to make our model focus on the rainfall change between adjacent time steps, learn a coherent temporal evolution of the TC rainfall, and have the capability of rainfall change awareness. This mechanism can reduce cumulative errors and ensure physical consistency, making the results more realistic and accurate.

- A multi-modal framework is built with several encoder modules to extract extensive information from various meteorological historical data and future predictions provided by NWP. This framework can represent the TC rainfall more comprehensively and provide a "bridge" between DL models and NWP models.

- We show that our model achieves the best performance compared to state-of-the-art (SOTA) DL precipitation forecasting methods and outperforms NWP methods from authoritative meteorological agencies like ECMWF.

## 2. Task Definition

Different from regular precipitation forecasting, where the predicted regions are fixed, our TC precipitation forecasting task focuses on the $10°$ by $10°$ rainfall field around the TC center, which moves with the TC. We denote the final output of our work as $\hat{Y} = \{\hat{y}_{n+1}, \hat{y}_{n+2}, \ldots, \hat{y}_{n+t}, \ldots, \hat{y}_{n+m}\}$, where $\hat{y}_{n+t}$ represents the rainfall prediction around the TC center $t$ time steps into the future. Here, $n$ represents the input time steps and $m$ represents the output time steps. The input data we used can be divided into two parts, as shown in the left part of Figure 2.(b): the first part is historical observation data ($X_{historical} = \{X_1^h, X_2^h, \ldots, X_t^h, \ldots, X_n^h\}$), where $X_t^h$ is the input data at the $t$ time step and it includes the rainfall value data, $\Delta$ Rainfall, the 2-Dimension environment data, and scalar TC variables, like TC intensity and track. The second part is the future prediction data from an NWP system ($X_{future} = \{X_1^f, X_2^f, \ldots, X_t^f, \ldots, X_m^f\}$), where $X_t^f$ denotes the prediction data at future $t$ time step. Overall, imagine a TC developing, just like shown in Figure 1.(c). We obtain all the input data $X = \{X_{historical}, X_{future}\}$ and then we input them to our model TCP-Diffusion. TCP-Diffusion will extract the TC information from the input data and make a rainfall prediction $\hat{Y}$. The aim is for $\hat{Y}$ to be as close as possible to the real TC rainfall ($Y = \{y_{n+1}, y_{n+2}, \ldots, y_{n+t}, \ldots, y_{n+m}\}$, where $y_{n+t}$ represents the real rainfall around the TC center at the future

$t$ time step), both in terms of the overall intensity and the spatial and temporal structure.

## 3. Model Design

TCP-Diffusion is a temporal DM-based method. The denoising process is performed by a neural network, which we call Environmentally-Aware 3DUNet (EA-3DUNet) (described in detail below). As shown in Figure 2.(b), it can use Historical $Data_{2d}$ Encoder, Historical $Data_{1d}$ Encoders and Future $Data_{2d}$ Encoder to extract TC rainfall information from rainfall related data, 2D environmental factor data, scalar TC variable and future prediction data. 3DUNet is designed to further extract spatiotemporal future features from the information provided by different encoders. The inherent abilities of DMs enables making predictions that have structure that resembles real rainfall data. The information from all the inputs helps the predicted rainfall values more closely match the real rainfall data. In the following section, we will introduce the details of TCP-Diffusion. We have released the code at https://github.com/Zjut-MultimediaPlus/TCP-Diffusion.

### 3.1. Adjacent Residual Prediction (ARP)

In the TC rainfall forecasting task, predicting the rainfall value directly is difficult because weather systems exhibit chaotic characteristics. Some NWP methods improve the accuracy and stability of forecasts by keeping the weather change as their prediction goal (Kalnay, 2003). Inspired by this idea, we find the capability of change awareness is helpful to our model, so we replace the direct prediction of rainfall value with predicting the adjacent residual $\Delta_x^t$ between rainfall at adjacent time steps ($\Delta$ Rainfall). We call this mechanism Adjacent Residual Prediction. $\Delta_x^t$ can be calculated as:

$$\Delta_x^t = X_{rain}^t - X_{rain}^{t-1} \qquad (2)$$

where $X_{rain}^t$ and $X_{rain}^{t-1}$ are the absolute rainfall data at $t$ time step and $t-1$ time step respectively. In our work, in addition to the absolute rainfall data, we add adjacent residual sequence $D_x = \{\Delta_x^1, \Delta_x^2, \ldots, \Delta_x^t, \ldots, \Delta_x^n\}$, where $\Delta_x^t$ is calculated by the Equation 2, which shown as the $\Delta$ Rainfall in Figure 2.(b). The direct output sequence of our DL model is $\tilde{D}_y = \{\hat{\Delta}_y^{n+1}, \hat{\Delta}_y^{n+2}, \ldots, \hat{\Delta}_y^{n+t}, \ldots, \hat{\Delta}_y^{n+m}\}$, where $\hat{\Delta}_y^{n+t}$ is the adjacent residual we predict between rainfall of time step $n+t-1$ and $n+t$. The final prediction $\hat{Y} = \{\hat{y}_{n+1}, \hat{y}_{n+2}, \ldots, \hat{y}_{n+t}, \ldots, \hat{y}_{n+m}\}$ can be computed as:

$$\hat{y}_{n+t} = X_{rain}^n + \sum_{z=1}^{t} \hat{\Delta}_y^{n+z} \qquad (3)$$

$\hat{Y}$ is obtained by the accumulation of the latest historical TC rainfall data $X_{rain}^n$ and the ARP $\hat{\Delta}_y$.

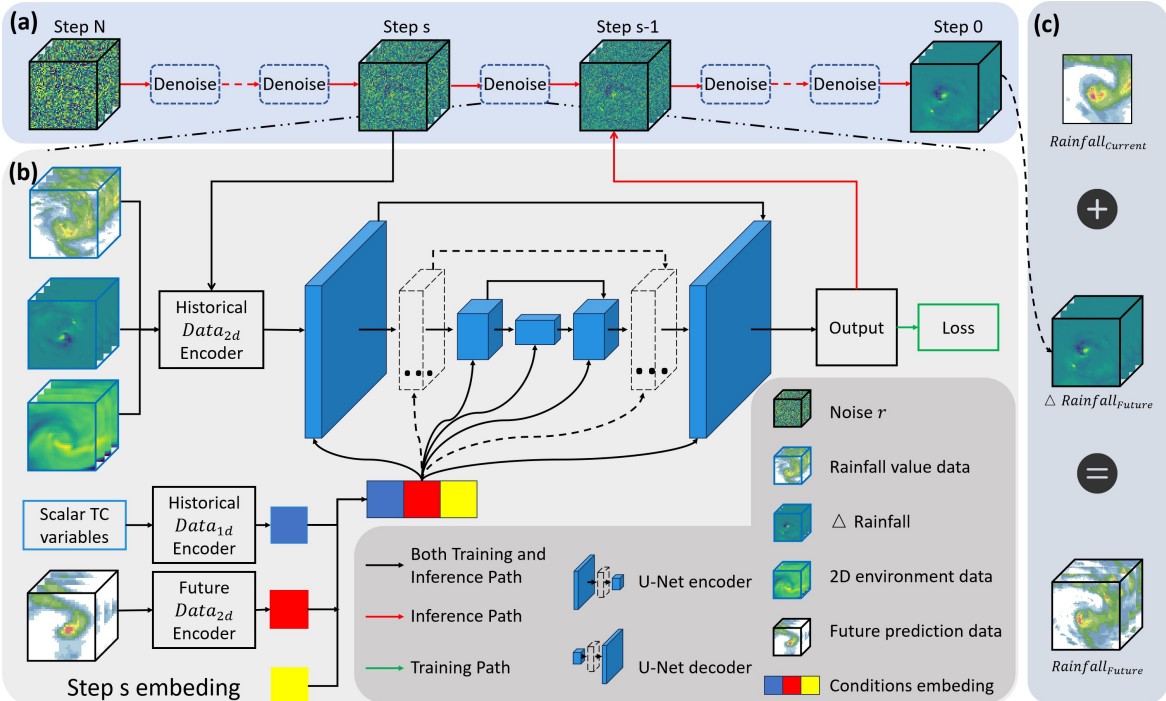

*Figure 2.* The framework of TCP-Diffusion. Sub-figure (a) shows the inference stage of TCP-Diffusion (black+red arrows). Our model will use $N$ loop denoising processes to obtain the future rainfall change $\Delta Rainfall_{Future}$ from a noise input. The sub-figure (b) shows the model structure of the denoising process and also shows the training stage of TCP-Diffusion (black+green arrows). In the training stage, the Output will be used to calculate the loss and update the parameters of TCP-Diffusion (green arrow). In the inference stage, the Output will be used to get the future rainfall change $\Delta Rainfall_{Future}$ with noise at Step $s-1$ (red arrow). The sub-figure (c) shows the process (Equation 1) of getting the final predicted TC rainfall $Rainfall_{Future}$ with the latest rainfall $Rainfall_{Current}$ and the future rainfall change $\Delta Rainfall_{Future}$.

### 3.2. Diffusion Model

The prediction process of DMs involves removing noise from a completely random noise $r$ step by step, shown in Figure 2.(a). Based on the DMs' generation process, our training goal is to teach our model to perform accurate denoising based on the specific step $s$ and the given information. Specifically, the training goal is to make the noise $\hat{r}_s$ predicted by our model as close as possible to the noise $r_s$ added to the target at the $s$-th step, $s \in \{1, 2, \ldots, N\}$. $N$ is a hyper-parameter.

DMs typically include two processes: a forward noising process and a reverse denoising process. The forward process has no trainable parameters, so it does not require training a DL model. According to the training goal mentioned above, we need to obtain the sequence $D_y^s$, where $D_y^s$ is $s$-th-step noised real adjacent residual $D_y = \{\Delta_y^{n+1}, \Delta_y^{n+2}, \ldots, \Delta_y^{n+t}, \ldots, \Delta_y^{n+m}\}$. Following (Ho et al., 2022), $D_y^s$ could be calculated using a random noise $r_s \sim \mathcal{N}(0, 1)$, step $s$, and the $D_y^0$ (the original data without noise). The forward process is defined as follows:

$$D_y^s = \sqrt{\bar{\alpha}_s} D_y^0 + \sqrt{1 - \bar{\alpha}_s} r_s \qquad (4)$$

where $\bar{\alpha}_s = \prod_{s=1}^{N}$, $\alpha_s = 1 - \beta_s$, $\beta_s \in (0, 1)$ is predefined by an incremental variance schedule.

For the denoising process, we build the EA-3DUNet, which is defined as follows:

$$\hat{r}_s = EA3DUNet(X_{history}, X_{future}, D_y^s, s, r) \qquad (5)$$

where $r$ is the added random noise, $EA3DUNet$ is the trainable model, and $\hat{r}_s$ is the predicted noise. Thus, our training goal can be defined as:

$$L(\theta) = \|r_s - \hat{r}_s\|_2 \qquad (6)$$

where $\theta$ represents all the trainable parameters in our model. $r_s$ is calculated by Equation 4.

For the inference phase, we consider a random noise $r$ as the $D_y^N$ (Ho et al., 2020) and use the following Equation 7 to denoise $D_y^N$ step by step to obtain the $\hat{D}_y^0$, which is equivalent to $\hat{D}_y$:

$$D_y^{s-1} = \frac{1}{\sqrt{\alpha_s}} \left( D_y^s - \frac{\beta_s}{\sqrt{1 - \bar{\alpha}_s}} \hat{r}_s \right) + \sigma_s \epsilon \qquad (7)$$

where $\hat{r}_s$ is the predicted noise added at $s$-th step, $\sigma_s$ is a variance hyperparameter. $\epsilon \sim \mathcal{N}(0, 1)$ is a random noise,

which plays a critical role in generating high-quality prediction in the inference phase (Ho et al., 2020).

Overall, the training phase (Figure 2.(b)) and inference phase (Figure 2.(a) and (c)) can be represented by the following pseudocode. Here, $q(\cdot)$ denotes the data distribution from which samples are drawn. Specifically, $q(D_y)$ represents the distribution of the target variable, while $q(X_{history})$ and $q(X_{future})$ represent the distributions of historical and future input features, respectively.

---

**Algorithm 1** Training

---

1: **repeat**
2:    $D_y^0 \sim q(D_y)$
3:    $s \sim \text{Uniform}(\{1, \ldots, N\})$
4:    $D_y^s = \sqrt{\bar{\alpha}_s} D_y^0 + \sqrt{1 - \bar{\alpha}_s} r$
5:    $X_{history} \sim q(X_{history})$
6:    $X_{future} \sim q(X_{future})$
7:    $r \sim \mathcal{N}(0, 1)$
8:    $\hat{r} = EA3DUnet(X_{history}, X_{future}, D_y^s, s, r)$
9:    Take gradient descent step on $\nabla_\theta \|r - \hat{r}\|^2$
10: **until** converged

---

**Algorithm 2** Inference

---

1: $D_y^N \sim \mathcal{N}(0, 1)$
2: $X_{history} \sim q(X_{history})$
3: $X_{future} \sim q(X_{future})$
4: **for** $s = N, \ldots, 1$ **do**
5:    $\epsilon \sim \mathcal{N}(0, 1)$ if $s > 1$, else $\epsilon = 0$
6:    $\hat{r} = EA3DUnet(X_{history}, X_{future}, \hat{D}_y^s, s, r)$
7:    $\hat{D}_y^{s-1} = \frac{1}{\sqrt{\alpha_s}}(\hat{D}_y^s - \frac{\beta_s}{\sqrt{1 - \bar{\alpha}_s}} \hat{r}) + \sigma_s \epsilon$
8: **end for**
9: Use Equation.3 to get $\hat{Y}$
10: **return** $\hat{Y}$

---

### 3.3. Environmentally-Aware 3DUNet (EA-3DUNet)

EA-3DUNet is the neural network component of TCP-Diffusion. It can extract various features from multi-modal data sources. Due to the differing characteristics and dimensions of these heterogeneous meteorological data, we need to build multiple encoders to embed these data. Therefore, we build the **Historical Data Encoder** for Rainfall Data and Environment Data, and the **Future Prediction Data Encoder** for future prediction data. To better extract temporal and spatial information from these data, we build **3DUNet** with a temporal and spatial attention mechanism. We will introduce all these modules in the following sections.

**Historical Data Encoder**   Rainfall Data $X_{rain}$, adjacent residual data $D_x$, ERA5 surface data $X_{SfEnv}$, and ERA5 pressure level data $X_{PlEnv}$ are two-dimensional (2D) data.

Considering that our data has a time dimension, we build a Historical $Data_{2d}$ Encoder with 3D Convolutional Neural Networks (CNNs) to encode all 2D data with time information first. To reduce computational resources, we use one module to encode these data, so we concatenate them to get $X_{his2D} = [X_{rain}, D_x, X_{SfEnv}, X_{PLEnv}, r_s]$. This process is defined as follows:

$$e_{his2D} = Conv3d(X_{his2D}, W_{his2D}) \tag{8}$$

where $W_{his2D}$ is the encoder parameters. $e_{his2D}$ represents the features encoded from $X_{his2D}$ and serves as the initial feature input to 3DUNet for further feature extraction.

The scalar TC variable $X_{Sc}$ does not have a 2D structure, so we build a Historical $Data_{1d}$ Encoder with Multilayer Perceptron (MLP) and Transformer (Vaswani et al., 2017) layers to encode $X_{Sc}$ and obtain the temporal information. The main process is as follows:

$$e_{mlp} = \phi(X_{Sc}, W_{mlp}) \tag{9}$$

$$e_{his1D} = \text{Transformer}(e_{mlp}, W_{atten}) \tag{10}$$

where $\phi$ is denoted as an MLP module, which is used for getting the features $e_{mlp}$ of different variables. $W_{mlp}$ is the parameters of $\phi$. The Transformer module is used for obtaining the temporal information of $X_{Sc}$. $W_{atten}$ is the parameter of Transformer and $e_{his1D}$ is the encoded features of $X_{Sc}$.

**Future Prediction Data Encoder**   Future prediction data $X_{future}$ contain future information and some physical information provided by NWP, which differs from Historical data. It is challenging for a single module to encode these data containing different types of information at the same time (Alzubaidi et al., 2021). Therefore, we build a Future $Data_{2d}$ Encoder to encode these data and try to obtain more specific features. We use a modified Resnet-18 (He et al., 2016) to obtain the vector. The main process of Future $Data_{2d}$ Encoder is as follows:

$$e_{future} = \text{Resnet}(X_{future}, W_{res18}) \tag{11}$$

where Resnet is the Resnet-18 model, $W_{res18}$ is the parameters, and $e_{future}$ is the condition extracted from $X_{future}$ and used for guiding our model to make a better prediction.

**3DUNet**   3DUNet (Çiçek et al., 2016) is the core component of EA-3DUNet and is a classical DL structure for tasks involving 2D data with time information. It usually includes three parts shown in Figure 2.(b): U-Net encoder ($U\text{-}Net_{en}$), U-Net decoder ($U\text{-}Net_{de}$), and the bottleneck between encoder and decoder. There are several modules in $U\text{-}Net_{en}$ and $U\text{-}Net_{de}$, called $module_{en}$ and $module_{de}$

respectively. In each module, we build different blocks (blue cuboids) to capture features from various aspects. Over all, the 3DUNet receives the feature map from the Historical $Data_{1d}$ Encoder and the condition information from Future $Data_{2d}$ Encoder. Additionally, step $s$ is also included in the condition information to inform the model how much noise to remove at the current stage. Then, the 3DUNet predicts the noise ($\hat{r}_s$) added at the step $s$. Finally, there are two pathes shown in Figure 2.(b). In the training stage (green arrow), we could calculate the loss between actual noise $r_s$ and $\hat{r}_s$ to optimize the parameters of our model. In the inference stage (red arrow), $\hat{r}_s$ could be used for obtaining $D_y^{s-1}$ using Equation 7. The main process of 3DUNet is shown in Appendix.B.

## 4. Dataset and Evaluation Methods

### 4.1. Dataset

We collected various TC rainfall-related data for our TCP-Diffusion model to comprehensively represent TC rainfall. A total of 1877 TCs spanning from 1980 to 2020 are collected, covering the six major ocean areas. These TC data are divided into three sets: training set (1751 TCs), validation set (87 TCs), and test set (126 TCs from 2018 to 2020). We divided the data into two parts: $X_{historical}$ and $X_{future}$. Details of the data we used are in Appendix.A.

### 4.2. Evaluation Metrics

**Equitable Threat Score (ETS)** The Equitable Threat Score (ETS) (Gandin & Murphy, 1992) is a metric used to evaluate the accuracy of precipitation forecasts. Compared to the Critical Success Index (CSI) (Schaefer, 1990), a more popular metric, it considers the effects of random chance, providing a more equitable assessment of a forecast model's performance (Manzato & Jolliffe, 2017). The definition of ETS is given in Appendix.C. We use this metric to show the performance for predicting light, medium, and heavy rainfall, setting thresholds 6 mm/3hr (ETS-6), 24 mm/3hr (ETS-24), and 60 mm/3hr (ETS-60) respectively.

**Total Precipitation Mean Absolute Error (TP$_{MAE}$)** : This is the absolute difference between real TC rainfall $Y$ and the predicted rainfall $\hat{Y}$ averaged over the $10°$ spatial domain, to show the prediction skill of different methods at predicting total TC rainfall.

## 5. Results

### 5.1. Comparison with State-of-the-art DL Methods

TCP-Diffusion is compared with a deterministic DL method: U-Net (Çiçek et al., 2016). We also compare our model with two generative DL methods developed for nowcasting pre-

| Model Name | ETS-6 ↑ | ETS-24 ↑ | ETS-60 ↑ | TP$_{MAE}$ ↓ |
|---|---|---|---|---|
| Persistence | 0.41640 | 0.14530 | 0.00564 | 0.44558 |
| U-Net | **0.44169** | 0.10587 | 0 | 0.47452 |
| PreDiff | 0.38453 | 0.11931 | 0.00430 | 0.53617 |
| NowcastNet | 0.42180 | 0.08990 | 0.00016 | 0.56954 |
| TCP-Diffusion | 0.43788 | **0.14703** | **0.00644** | **0.42344** |

↑↓ ↑ Higher is better. ↓ Lower is better.
* Bold values are the best. Underlined values are the second best.

*Table 1.* Comparison with other SOTA DL methods. Equitable Threat Score (ETS) is used to evaluate the prediction skill of each model. ETS-6, ETS-24, and ETS-60 are used to show the prediction skill for light ($>6$ mm/3hr), medium ($>24$ mm/3hr), and high ($>60$ mm/3hr) rainfall respectively. TP$_{MAE}$ shows the total precipitation forecasting skill of different methods and, with unit mm/3hr. The results here represent the overall performance of each model for lead times of 3, 6, 9, and 12 hours.

cipitation: PreDiff, which is DM-based (Gao et al., 2024), and NowcastNet (Zhang et al., 2023). We also compare TCP-Diffusion with a basic baseline, persistence forecasting, using the last observed rainfall field for each future prediction (Panofsky, 1963). Since TCP-Diffusion is a probabilistic prediction model, to ensure the stability of the results presented, we conducted eight tests on the test set. For each test, the evaluation metrics were calculated, and the final results were obtained by averaging the metrics across these eight tests. Additionally, the individual results from each test are provided in Table 5 in Appendix.D.2.

**Qualitative Analysis.** Figure 3 shows an example of predictions from TCP-Diffusion and other DL methods for a forecast of TC Dumazile. Comparing with the deterministic U-Net rainfall prediction model, the DM-based models (Predif and TCP-Diffusion) can give more realistic rainfall predictions with fine spatial detail, such as the shape of the rain band. U-Net and NowcastNet produce predictions that are too spatially smooth. This is one of the reasons we chose DMs as the basis for our TC rainfall prediction model. When comparing our TCP-Diffusion with PreDiff in this example, our model provides more accurate results and does better at predicting the increasing trend of precipitation intensity with time, because the ARP mechanism gives our model the capability to track the rainfall change. We show two samples from TCP-Diffusion, generated using different noise inputs. We also show all 8 samples from TCP-Diffusion in Figure 9 in Appendix.D. Overall, they have similar large-scale rainfall trends and structures. This means that the information extracted from other input data, such as historical TC rainfall and 2D environmental variables, can guide our model to make reasonable and realistic forecasts. Overall, these results indicate that TCP-Diffusion can produce rainfall predictions that have realistic spatial and temporal structure, performing better in this example than the other DL methods.

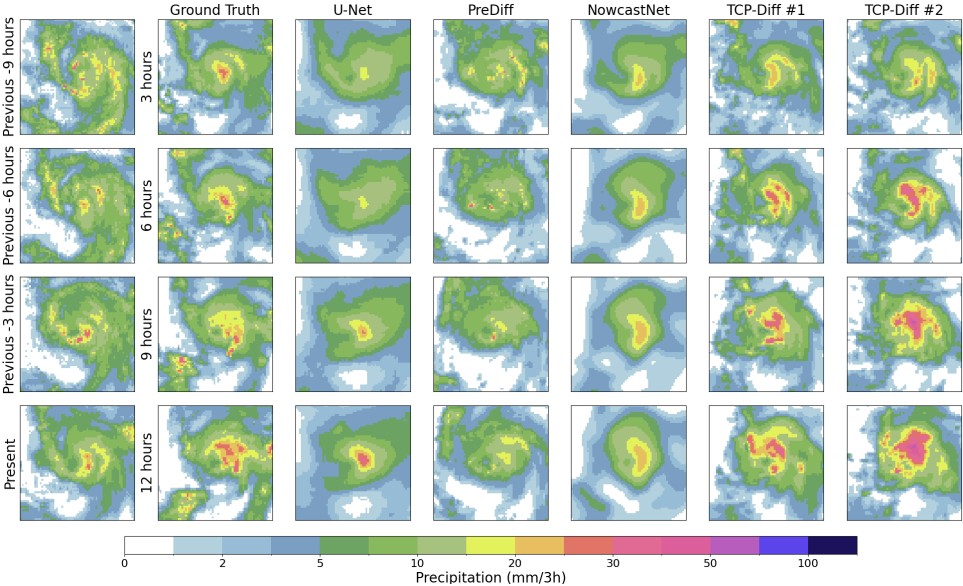

*Figure 3.* The prediction results of different DL methods on TC Dumazile in the North Indian Ocean at 03/03/2018 06:00. The first column is the previous 4 timesteps of rainfall data used as input. The second column is the future TC rainfall we want to predict. Each subsequent column from the left shows the predictions from a different DL method. "TCP-Diff #1" and "TCP-Diff #2" are two samples from our TCP-Diffusion model with different initial random noise $r$.

**Quantitative Analysis.** To further demonstrate the effectiveness of our method, we calculate and compare various metrics on the entire test set, as shown in Table 1. We also show the performance of TCP-Diffusion compared with other methods at lead times of 3, 6, 9, and 12 hours separately in Figure 8 in Appendix.D. The lead time indicates the amount of time between the issuance of a forecast and the actual occurrence of the weather event. For light rainfall prediction (ETS-6), U-Net achieves the best performance, with TCP-Diffusion ranking second. However, U-Net performs poorly for moderate and heavy rainfall predictions. For higher rainfall thresholds and $TP_{MAE}$, our TCP-Diffusion method achieves the best performance. Especially in heavy rainfall prediction, TCP-Diffusion shows a substantial improvement compared to other DL models. It is also the only model that performs better than Persistence forecasting, indicating the value of designing a specialised system for the task of TC rainfall prediction. We also show some analysis about why Persistence forecasting performs better than the other three DL models in Appendix.D.

**Analysis of TC Rainfall Frequency Distribution.** Figure 4 shows histograms of rainfall rates for observations and forecasts by each DL method. We find that the performance of DM-based models, PreDiff and TCP-Diffusion, is much better than that of non-DM-based models, especially in producing a realistic frequency of the heaviest rainfall rates. Our method predicts a slightly higher frequency of the heaviest precipitation intensities than is observed, while PreDiff

predicts a slightly lower precipitation intensity. However, for more moderate rainfall intensities, TCP-Diffusion has a more realistic frequency than PreDiff, as shown in the circled region in Figure 4. We also present the distribution of rainfall prediction from TCP-Diffusion and compared methods at each lead-time in Figure 10 in Appendix.D.

**Analysis of TC Rainfall's Radially Averaged Power Spectral Density** We use the Radially Averaged Power Spectral Density (RAPSD) to help demonstrate that our model better captures the spatial structure of large-scale TC rainfall systems, as shown in Figure 5. If the power spectral density curve predicted by the model is closer to the ground truth MSWEP at a given spatial frequency, it indicates that the model captures the rainfall variability at that spatial scale more effectively. The low wavenumber region corresponds to large-scale weather systems, such as fronts and tropical cyclones. As observed in Figure 5, the power spectral density of TCP-Diffusion is closer to MSWEP compared to PreDiff in the low wavenumber region, indicating that our method achieves better capture of rainfall characteristics at the TC scale. We also show the RAPSD of rainfall prediction from TCP-Diffusion and compared methods at each lead-time in Figure 11 in Appendix.D.

### 5.2. Comparison with NWP Methods

We also compare our model with two state-of-the-art NWP methods. We select the precipitation forecast data in THORPEX Interactive Grand Global Ensemble (TIGGE) provided

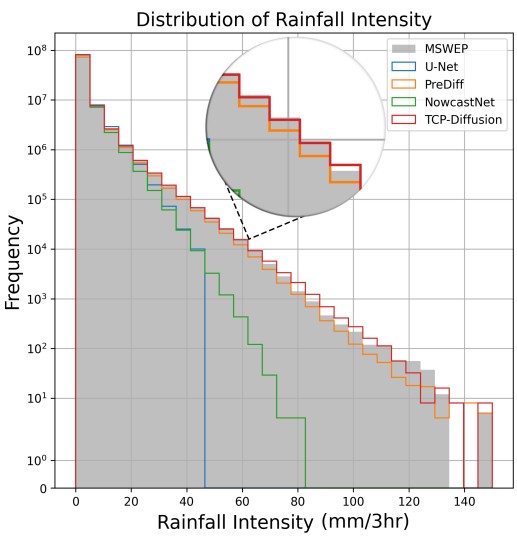

*Figure 4.* TC rainfall frequency distributions of different DL forecasting methods. The grey histogram is the distribution of MSWEP observations. The coloured lines show histograms of rainfall rates from different DL forecasting methods. The circular magnified region more clearly shows differences between these methods in the indicated span of rainfall intensity. Note the logarithmic vertical axis.

| Model Name | ETS-6 ↑ | ETS-24 ↑ | ETS-60 ↑ | $TP_{MAE}$↓ |
|---|---|---|---|---|
| ERA5-IFS | 0.20172 | 0.01646 | 0 | 0.51134 |
| ECMWF-IFS | 0.30193 | 0.08348 | 0.00255 | 0.50661 |
| TCP-Diffusion | **0.41238** | **0.12790** | **0.00444** | **0.47446** |

*Table 2.* Comparison with the ERA5-IFS and ECMWF-IFS NWP methods. The forecast diagnostics are the same as in table 1.

by ECMWF (ECMWF, 2006), denoted as ECMWF-IFS. This has higher skill than the ERA5-IFS (ECMWF, 2017). However, the spatial resolution of ERA5-IFS results is lower than that of ECMWF-IFS results, which means the costs of getting ERA5-IFS results is lower than that of ECMWF-IFS results and makes ERA5-IFS results more practical to use as the future prediction data $X_{future}$ in training. As ECMWF-IFS forecasts in the TIGGE archive are for 6-hourly rather than 3-hourly precipitation, we sum 3-hourly TCP-Diffusion forecasts within each 6-hour interval. Additionally, ECMWF-IFS forecasts start at 12 am and 12 pm UTC each day, so we use TCP-Diffusion to forecast TC rainfall from the same start times. Thus, the test samples in this table are different from those in Table 1.

As shown in Table 2, our method achieves better ETS for all rainfall thresholds and better $TP_{MAE}$ than the ECMWF-IFS. This means that low-cost, low-quality NWP methods (ERA5-IFS), when augmented by DL techniques, have the potential to surpass the performance of high-cost, high-quality NWP approaches (ECMWF-IFS).

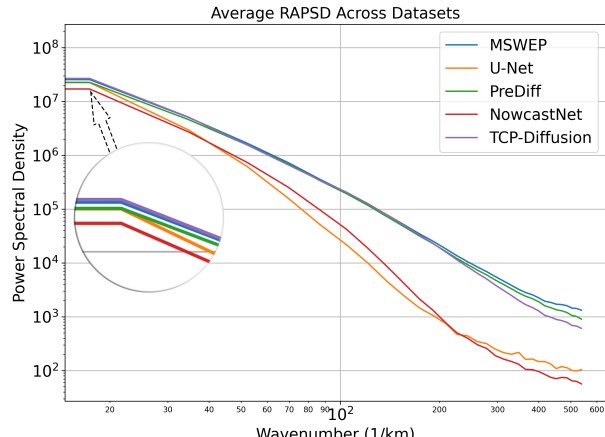

*Figure 5.* Radially Averaged Power Spectral Density (RAPSD) analysis across datasets. The plots compare the power spectral density of predicted rainfall using different models over a range of wavenumber. In the low wavenumber region, higher power spectral density represents that the displayed data belongs to large-scale weather systems, such as fronts and tropical cyclones. The power spectral density of TCP-Diffusion in the low wavenumber region is closer to the ground truth rainfall (MSWEP), indicating that our method achieves better capture of rainfall characteristics at the TC scale. Note the logarithmic vertical axis.

### 5.3. Ablation Studies

To demonstrate the effectiveness of individual components of our model, an ablation study is conducted. As shown in Table 3, comparing the results from the model with and without **ARP** in the first two rows, we find that changing the regular rainfall value prediction to the adjacent residual prediction results in an improvement in ETS of about 0.1–15.0%. This indicates that predicting the adjacent residual reduces cumulative errors. Then, due to the rich and important TC environment information from multi-modal meteorological data, the model with **M** and **ARP** (third row) performs better than the model with only **ARP**, showing an improvement of about 5.6–13.0% in ETS and a reduction in

| ARP | M | F | ETS-6 ↑ | ETS-24 ↑ | ETS-60 ↑ | $TP_{MAE}$↓ |
|---|---|---|---|---|---|---|
| | | | 0.39109 | 0.12474 | 0.00453 | 0.50109 |
| ✓ | | | 0.40634 | 0.13014 | 0.00521 | 0.50043 |
| ✓ | ✓ | | 0.42926 | 0.14253 | 0.00589 | 0.44632 |
| ✓ | ✓ | ✓ | **0.43788** | **0.14703** | **0.00644** | **0.42344** |

*Table 3.* Ablation studies. **ARP** means the contribution that we change the regular rainfall value prediction to the adjacent residual prediction. **M** means we not only consider the information from TC rainfall but also consider the TC-related environment information extracted from multi-modal meteorological data. **F** means we combine our DL model with the NWP method by using the predictions provided by NWP to guide the prediction process of our method.

$TP_{MAE}$. This demonstrates the significance of building different encoder modules to extract TC rainfall-related information, addressing the deficiency of information provided only by the TC rainfall field. Finally, our complete model TCP-Diffusion including NWP forecast inputs (fourth row) achieves a 2.0–9.3% improvement in ETS over the model with only **M** and **ARP**. This demonstrates that using forecasts provided by NWP to guide DL prediction can enhance skill. (Though note that the scores for our model without using future forecast data still improve upon those of other benchmarks for ETS at high thresholds and $TP_{MAE}$ shown in table 1, showing that the method can improve skill even when future forecast data is unavailable.) Overall, due to the contributions of **ARP**, **M**, and **F**, our final model TCP-Diffusion achieves an improvement in ETS scores of about 11.9–42.1% over the baseline original spatio-temporal diffusion model across the different metrics.

# 6. Conclusion

In this paper, we propose the TCP-Diffusion model for forecasting TC precipitation in the region around a given track location, applicable anywhere in the world. TCP-Diffusion uses the ARP mechanism to predict changes in rainfall and a framework with multiple encoders to extract information from numerous relevant variables, including historical TC rainfall data, TC-related environmental data, and forecast data from NWP models. These guide and control our model to make better predictions. TCP-Diffusion produces precipitation predictions with realistic spatial variability. It achieves higher skill scores than existing rainfall forecasting methods and a leading NWP system, the ECMWF IFS. In the future, our model can be applied as a component of a forecasting system that includes track and intensity forecasts to produce overall predictions of TC properties.

## Acknowledgements

This work is partially supported by the Zhejiang Provincial Natural Science Foundation of China under Grant No. LRG25F020002 and No. LR21F020002, Natural Science Foundation of China under Grant No. 62202429, U24A20221 and Zhejiang Provincial Natural Science Foundation of China under Grant No. LY23F020024. Peter A. G. Watson was supported by a NERC Independent Research Fellowship (Grant No. NE/S014713/1).

## Impact Statement

This work aims to advance the accuracy and reliability of tropical cyclone (TC) rainfall forecasting, addressing a critical gap in weather prediction. By leveraging deep learning and integrating meteorological factors with numerical weather prediction (NWP) models, the proposed TCP-Diffusion method significantly improves TC rainfall forecasts while reducing cumulative errors and ensuring physical consistency. These advancements have the potential to enhance disaster preparedness and mitigation strategies, provide actionable insights for climate resilience, and improve the overall efficiency and effectiveness of global weather prediction systems.

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

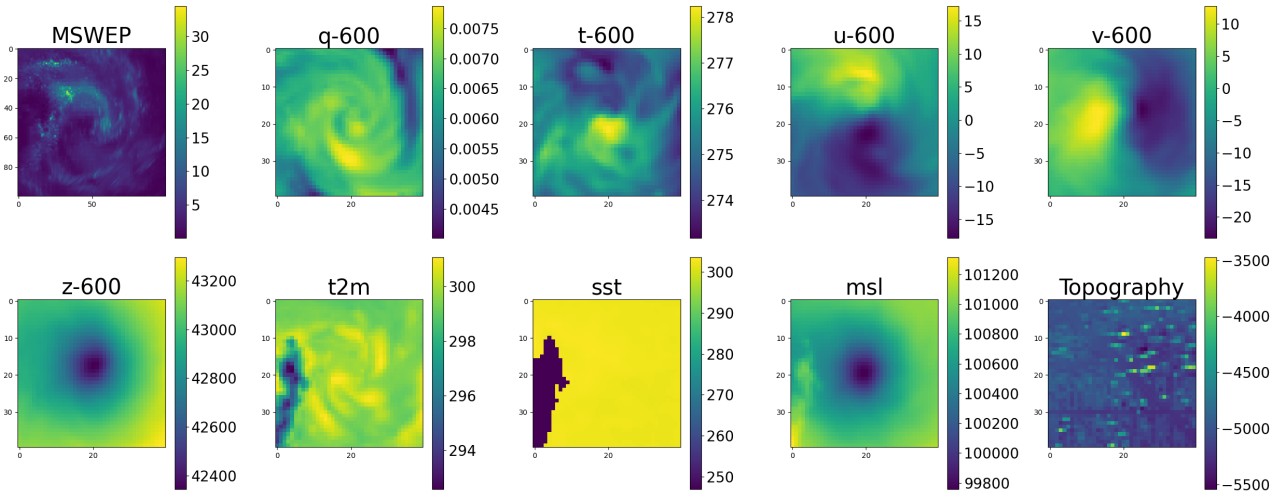

*Figure 6.* The visualization presents part of the historical data on TC Dumazile in North Indian Ocean at 03/03/2018 06:00. MSWEP is the rainfall values. q-600, t-600, u-600, v-600, z-600 represent the Specific Humidity, Temperature, U Wind, V Wind, and Geopotential Height at the 600 hPa pressure level respectively. t2m, sst, msl and topography represent the 2m Temperature, Sea Surface Temperature, Mean Sea Level Pressure, and Topography respectively around the TC center.

## A. The Details of Data

We divided the data into two parts: $X_{historical}$ and $X_{future}$. We collected a total of 1877 TCs spanning from 1980 to 2020 contained in the International Best Track Archive for Climate Stewardship (IBTrACS) dataset (Knapp et al., 2010), covering the six major ocean areas. These TC data are divided into three sets: training set, validation set, and test set. The test set contains 126 TCs from 2018 to 2020. For the other two datasets, we randomly selected 95% of the TCs from 1980 to 2018 as the training set (1751 TCs) and 5% as the validation set (87 TCs). Due to the time cost of DMs inference, we use approximately 5% of the data to validate and determine our best checkpoint in the training run. Some of the $X_{historical}$ are visualized in Figure 6 for one TC at one time and some of the $X_{future}$ are visualized in Figure 7.

### A.1. Historical Data

**Rainfall Data**  The rainfall dataset that we use ($X_{rain}$) is the Multi-Source Weighted-Ensemble Precipitation (MSWEP)(Beck et al., 2019) global precipitation product. This has a 3-hourly $0.1°$ resolution available from 1979 to about 3 hours from real-time. We focus only on the rainfall field around the TC center, so we crop and obtain rectangular data covering a $10°$ by $10°$ region around the TC center.

**Environment Data**  We also collect critical meteorological variables including surface data and pressure level data (200 hPa, 600 hPa, 850 hPa, and 925 hPa) from ERA5 (ECMWF, 2017), and TC attributes data. The surface data, denoted as $X_{SfEnv}$, include 2m temperature, sea surface temperature, mean sea level pressure, and topography. ERA5 pressure level data, denoted as $X_{PlEnv}$, consists of 5 variables: temperature, specific humidity, U-component of wind, V-component of wind, and geopotential height. We perform the same operations as with $X_{rain}$, focusing on the $10°$ by $10°$ region around the TC center.

We also use several scalar TC variables, which we call $X_{Sc}$ and includes TC intensity, movement velocity, the month and the track location. $X_{Sc}$ is collected and calculated from the IBTrACS dataset.

### A.2. Future Prediction Data.

It is necessary for DL models to better understand the physical processes of TC rainfall. However, incorporating the physical mechanisms of TC rainfall development into DL model design is challenging. Physically-based NWP methods can predict TC rainfall using various equations, such as Dynamical Equations, Thermodynamic Equations, Moisture Equations, and

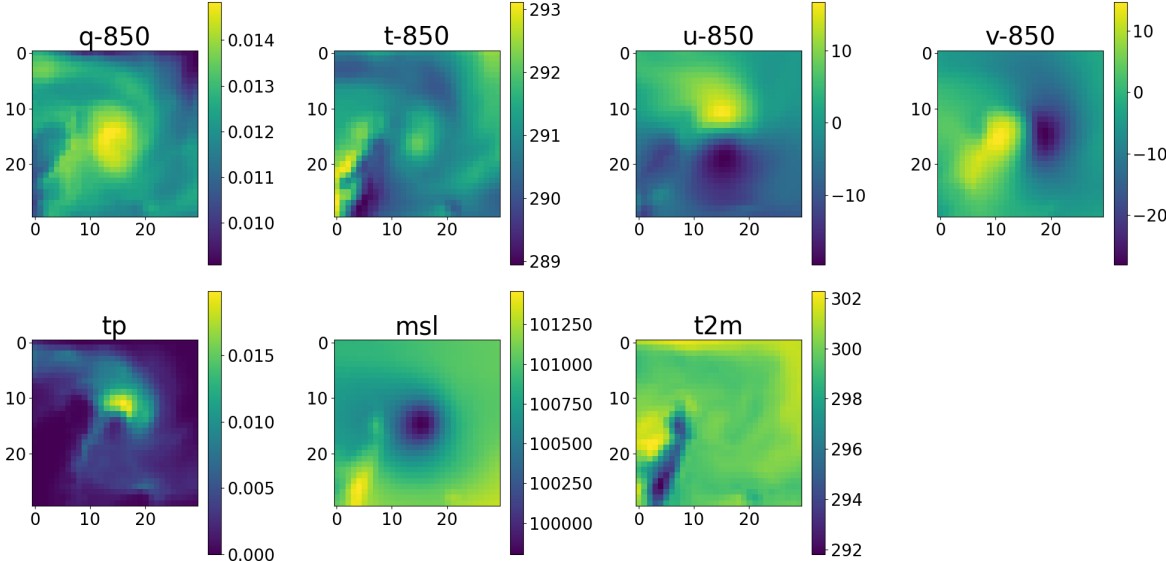

*Figure 7.* The visualization of the Future Prediction Data for the TC Dumazile in the North Indian Ocean at 03/03/2018 06:00. These data are all collected from ERA5. The q-850, t-850, u-850, and v-850 represent the forecast data for Specific Humidity, Temperature, U Wind, and V Wind at 850 hPa pressure level respectively. The tp, msl, and t2m represent Total Precipitation, Mean Sea Level Pressure, and 2m Temperature, respectively.

Radiative Transfer Equations, based on physical mechanisms. The prediction data are thus based on explicit physical laws. DL models are good at extracting information from data. Therefore, we also consider $X_{future}$ as input data for our model, providing future information based on physical laws to some extent. All $X_{future}$ we used is provided by the ensemble mean of the ERA5 (ERA5-IFS) (ECMWF, 2017). Here, we opted for the low-resolution ERA5-IFS dataset over the high-resolution ECMWF-IFS (ECMWF, 2006) for our training data due to ERA5-IFS's broader temporal coverage and higher temporal resolution. Additionally, the cost of generating ERA5-IFS is lower owing to its coarser spatial resolution. Moreover, our results reported in the main text showed that incorporating ERA5-IFS enhances the predictive performance of our model, surpassing that achieved with the high-quality ECMWF-IFS. There are many variables at the 200 or 850 hPa pressure levels, such as temperature, specific humidity, U-component of wind, and V-component of wind. Total precipitation, 2m temperature, and mean sea level pressure are also collected.

## B. Development of the model

### B.1. The Setting of Hyper-parameters

In designing our method, we established consistent hyperparameters to enable greater flexibility for other users. This allows users to apply TCP-Diffusion to similar tasks and tailor the model by adjusting specific hyperparameters according to their needs. The time steps for input data $n$ and future prediction $m$ are both set to 4, meaning our model extracts information from 12 hours of historical and ERA5 prediction data to predict accurate future 12-hour TC rainfall data. The denoising step $N$ is set to 200, but users can modify it to control the model inference time. For $\beta_s$, we use the Cosine Schedule to set it: $\beta_s = 1 - cos(\pi s/(N-1))$, which controls the level of noise addition at the specific step $s$. The $\sigma_s$ in Equation 7 in the main paper is related to $\beta_s$ and is set as $\sqrt{\beta_s}$. The count of modules $K$ in $U\text{-}Net_{en}$ and $U\text{-}Net_{de}$ is set as 4. People can use it to adjust the size of our model.

### B.2. Hardware Details

The experiments in this work were conducted using an NVIDIA A100 GPU with 256GB of RAM. The deep learning models were developed and trained using PyTorch (version 1.10) as the primary framework. The training process took approximately 72 hours to achieve the optimal checkpoint. During inference, each sample takes around 1.253 seconds to process. We also compare different methods' training and inference time-cost in Table 4. In addition, the environment included essential Python libraries such as NumPy (version 1.21), SciPy (version 1.7), and CUDA Toolkit (version 11.4) for

|                    | U-Net  | PreDiff | NowcastNet | TCP-Diffusion | ECMWF-IFS |
|--------------------|--------|---------|------------|---------------|-----------|
| Training (h)       | 5      | 67      | 52         | 72            | /         |
| Inference (s/sample) | 0.0062 | 9.401   | 0.0052     | 1.253         | 30min-2h  |

*Table 4.* Training and inference time-cost of TCP-Diffusion and compared methods.

GPU acceleration.

### B.3. The Detail of 3DUNet

3DUNet (Çiçek et al., 2016) is the core component of EA-3DUNet and is a classical DL structure for tasks involving 2D data with time information. It usually includes three parts shown in Figure 2.(b): U-Net encoder ($U\text{-}Net_{en}$), U-Net decoder ($U\text{-}Net_{de}$), and the bottleneck between encoder and decoder. There are several modules in $U\text{-}Net_{en}$ and $U\text{-}Net_{de}$, called $module_{en}$ and $module_{de}$ respectively. In each module, we build different blocks (blue cuboids) to capture features from various aspects.

Specifically, in the encoder module ($module_{en}$), shown as the blue cuboids in $U\text{-}Net_{en}$ in Figure 2, we stack two CNN blocks, one spatial attention (SA) block, one temporal attention (TA) block, and a down-sampling block. The network progressively performs downsampling by stacking several $module_{en}$. Each $module_{en}$ receives the feature map from the previous module and also the conditions $e_{his1D}$ and $e_{future}$ from the Historical $Data_{1d}$ Encoder and Future $Data_{2d}$ Encoder, respectively. Additionally, step $s$ is also received to inform the model how much noise to remove at the current stage. We concatenate these conditions to form the final condition $Cond = [e_{his1D}, e_{future}, s]$. The main process of $U\text{-}Net_{en}$ is as follows:

$$e_{en}^i = \begin{cases} module_{en}^i(e_{his2D}, Cond, W_{en_i}) & \text{if } i = 1, \\ module_{en}^i(e_{en}^{i-1}, Cond, W_{en_i}) & \text{if } i > 1. \end{cases} \tag{12}$$

where $module_{en}^i$ is the depth-$i$ module in $U\text{-}Net_{en}$, $i \in \{1, 2, \ldots, K\}$. $K$ is a hyper-parameter. $W_{en_i}$ is the parameters of the $module_{en}^i$. $module_{en}^1$ (the first $module_{en}$) receives $e_{his2D}$ from the historical data encoder. Subsequent $module_{en}^i$ ($i > 1$) receive $e_{en}^{i-1}$ from $module_{en}^{i-1}$.

The module between $U\text{-}Net_{en}$ and $U\text{-}Net_{de}$ is the Bottleneck. The Bottleneck also contains two CNN blocks, one SA block, and one TA block. The main process is as follows:

$$e_{mid} = Bottleneck(e_{en}^K, Cond, W_{neck}) \tag{13}$$

where $e_{mid}$ is the output of the Bottleneck and $W_{neck}$ is the parameter of Bottleneck.

For $U\text{-}Net_{de}$, its structure is similar to that of $U\text{-}Net_{en}$. They both include $K$ modules with similar architecture. In $module_{de}$, there are two CNN blocks, one SA block, one TA block and the final up-sampling block. There are also skip connections between $U\text{-}Net_{en}$ and $U\text{-}Net_{de}$. Thus, the definition of $U\text{-}Net_{de}$ is as follows:

$$e_{de}^i = \begin{cases} module_{de}^i(e_{mid}, Cond, e_{en}^i, W_{de_i}) & \text{if } i = K, \\ module_{de}^i(e_{de}^{i+1}, Cond, , e_{en}^i, W_{de_i}) & \text{if } i < K. \end{cases} \tag{14}$$

where $module_{de}^i$ is the depth-$i$ module in $U\text{-}Net_{de}$, $i \in \{K, \ldots, 2, 1\}$. $W_{de_i}$ is the parameters of the $module_{de}^i$. If $module_{de}^i$ is the deepest one ($i = K$), $module_{de}^K$ receives the $e_{mid}$ from the Bottleneck. if $module_{de}^i$ is not the deepest one ($i < K$), $module_{de}^i$ receives the $e_{de}^{i+1}$ from $module_{de}^{i+1}$. if $i = 1$, $e_{de}^i$ is the final output of EA-3DUNet, which means $\hat{r}_s = e_{de}^1$. Then, we could calculate the loss between $r_s$ and $\hat{r}_s$ to optimize the parameters of our model.

## C. The Definition of Metrics

### C.1. Equitable Threat Score (ETS)

The Equitable Threat Score (ETS) (Gandin & Murphy, 1992) is a statistical measure used to evaluate the accuracy of precipitation forecasts. Compared to the more popular metric, the Critical Success Index (CSI)(Schaefer, 1990), it provides a more robust assessment of a forecast model's performance (Manzato & Jolliffe, 2017). The definition of ETS is as follows:

$$R(a) = \frac{(N_A + N_B)(N_A + N_C)}{N_A + N_B + N_C + N_D} \tag{15}$$

$$ETS = \frac{N_A - R(a)}{N_A + N_B + N_C - R(a)} \tag{16}$$

where $N_A$, $N_B$, $N_C$, and $N_D$ represent the number of correctly predicted precipitation events, the number of false alarms, the number of missed precipitation events, and the number of correctly predicted no-precipitation events respectively. $R(a)$ denotes the expected number of correct forecasts due to random chance.

In addition, we also use this metric to show the performance of different methods for light, moderate, and heavy rainfall prediction. Thus, we set thresholds 6 mm/3hr (ETS-6), 24 mm/3hr (ETS-24), and 60 mm/3hr (ETS-60) to show the prediction skill for each rainfall category respectively. We need to do some processing before we calculate the ETS, which is shown as follows:

$$\hat{Y}_T = \begin{cases} 1 & \text{if } \hat{Y} \geq T, \\ 0 & \text{if } \hat{Y} < T. \end{cases} \tag{17}$$

$$Y_T = \begin{cases} 1 & \text{if } Y \geq T, \\ 0 & \text{if } Y < T. \end{cases} \tag{18}$$

where $T \in \{6, 24, 60\}$ is the rainfall intensity threshold. Then we use $\hat{Y}_T$ and $Y_T$ to calculate ETS-$T$ via Equations 15 and 16.

### C.2. Total Precipitation Mean Absolute Error (TP$_{MAE}$)

In addition to comparing the prediction skills of different methods in light, moderate, and heavy rainfall, we also want to evaluate the performance of total precipitation forecasting in the region covered by TC. This is a critical index for representing the intensity of TC rainfall. Therefore, we use Total Precipitation Mean Absolute Error (TP$_{MAE}$) to show the prediction skill of different methods in total rainfall prediction. The definition of TP$_{MAE}$ is as follows:

$$TP_{\text{MAE}} = \frac{\sum_{i=1}^{h} \sum_{j=1}^{w} \sum_{t=1}^{m} \left| Y_{ijt} - \hat{Y}_{ijt} \right|}{hwm} \tag{19}$$

where $h$ and $w$ represent the height and the width of the TC rainfall data respectively and $m$ represents the total number of time steps that we want to predict. $Y_{ijt}$ and $\hat{Y}_{ijt}$ are the observed and predicted rainfall at the given grid point and time respectively.

## D. Extended Experiments

### D.1. Results at Different Lead-times

In the main text, we have shown the overall performance of our model compared to other models. Now, we will demonstrate and compare their predictive performance at each lead time point—3-hour, 6-hour, 9-hour, and 12-hour. As illustrated in Figure 8, we evaluate the performance of each method using the metrics ETS-6, ETS-24, ETS-60, and TP$_{MSE}$. For ETS-6, the Unet model achieves the best performance, with TCP-Diffusion ranked second. However, for the remaining metrics, our method consistently outperforms the others. Notably, the ETS-60 results indicate that non-diffusion-based models do not perform well. Overall, TCP-Diffusion not only excels in overall performance but also demonstrates strong predictive accuracy at each lead time point. Besides, as the prediction lead time increases, the performance of all deep learning methods declines to varying degrees. In the future, extending the duration of rainfall prediction and improving its accuracy will be the focus of subsequent research efforts.

### D.2. Results of Multiple Tests

TCP-Diffusion is a probabilistic prediction model, which generates different results for different noise inputs. To ensure the reproducibility of the metric results for our model presented in the main text, we produced 8 separate samples for each TC case in the test set, with each test using independent random noise drawn from a normal distribution in the diffusion process. As a result, we obtained 8 independent prediction outcomes. After calculating the metrics for these outcomes, the results are shown in Table 5. From the experimental results, it can be observed that the performance of our model across multiple tests remains relatively stable on these metrics, with low standard deviations.

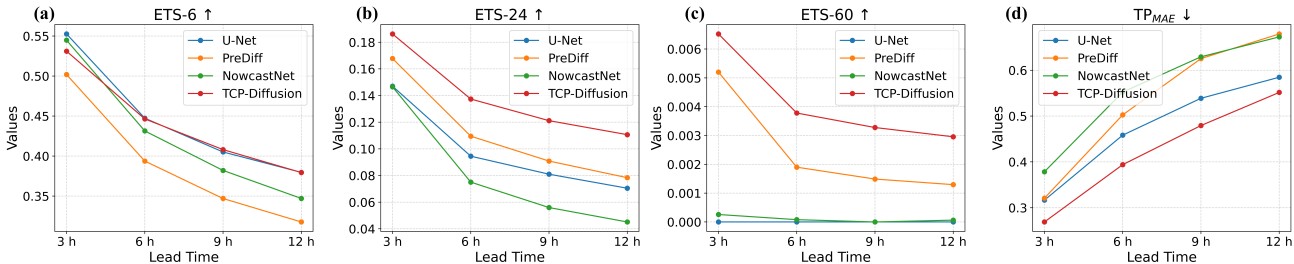

*Figure 8.* Comparison with SOTA deep learning methods for lead times of 3 h, 6 h, 9 h, and 12 h on ETS-6 (a), ETS-24 (b), ETS-60 (c), and TP$_{MSE}$ (d).

| Sample/Statistic | ETS-6 ↑ | ETS-24 ↑ | ETS-60 ↑ | TP$_{MAE}$↓ |
|---|---|---|---|---|
| TCP-Diffusion#1 | 0.43862 | 0.14848 | 0.00726 | 0.42403 |
| TCP-Diffusion#2 | 0.43867 | 0.14698 | 0.00619 | 0.41968 |
| TCP-Diffusion#3 | 0.43782 | 0.14735 | 0.00674 | 0.42340 |
| TCP-Diffusion#4 | 0.43735 | 0.14651 | 0.00646 | 0.42527 |
| TCP-Diffusion#5 | 0.43860 | 0.14802 | 0.00632 | 0.42131 |
| TCP-Diffusion#6 | 0.43715 | 0.14621 | 0.00633 | 0.42406 |
| TCP-Diffusion#7 | 0.43716 | 0.14591 | 0.00601 | 0.42265 |
| TCP-Diffusion#8 | 0.43767 | 0.14678 | 0.00623 | 0.42713 |
| TCP-Diffusion-mean | 0.43788 | 0.14703 | 0.00644 | 0.42344 |
| TCP-Diffusion-std | 0.00066 | 0.00088 | 0.00039 | 0.00229 |

*Table 5.* Performance metrics of TCP-Diffusion based on 8 independent tests, where TCR-Diffusion#1 to TCR-Diffusion#8 represent the results of the 1st to 8th predictions. The rows TCR-Diffusion-mean and TCR-Diffusion-std summarize the mean and standard deviation of these tests, respectively.

We also visualized the results of each prediction, as shown in Figure 9. The eight samples exhibit an overall trend consistent with the ground truth, with an enhancement in rainfall intensity. This demonstrates the effectiveness of our APR mechanism, showing that our model can accurately perceive future changes in rainfall trends and provide predictions aligned with these changes. Besides, the eight samples display differences in rainfall patterns and intensity, effectively simulating the chaotic nature of tropical cyclone rainfall. These multiple possible predictions are valuable for forecasters to determine the uncertainty range of the future TC rainfall. This approach aligns with the ensemble forecasting method in meteorology, which is widely used to address chaotic weather systems.

### D.3. Analysis of Persistence's Good Performance

There is an interesting phenomenon: Persistence forecasting performs better than the other DL methods except TCP-Diffusion for most evaluation metrics, as shown in Figure 1. There may be two reasons for this situation. On the one hand, these DL methods are designed for traditional rainfall prediction tasks, focusing only on a fixed region, meaning the observation window does not change with the movement of the rain band. Our TC rainfall prediction task focuses on the precipitation in the region around the TC center, and the observation window changes with the movement of the TC. There are some differences between these two tasks. Therefore, some novel ideas proposed for traditional rainfall prediction tasks may not be suitable for TC rainfall prediction. This means that simply applying previous rainfall forecasting methods to TC rainfall forecasting is insufficient, highlighting the value of proposing the TCP-Diffusion model, which is designed with consideration of the special features of TC. On the other hand, because we keep the observation window always centered around the TC, there is always rainfall in the center of the observation window, and the rainfall intensity usually does not change much over several hours. The rainfall is also primarily concentrated in the central region. Therefore, when calculating the metrics, this makes Persistence forecasting a strong baseline.

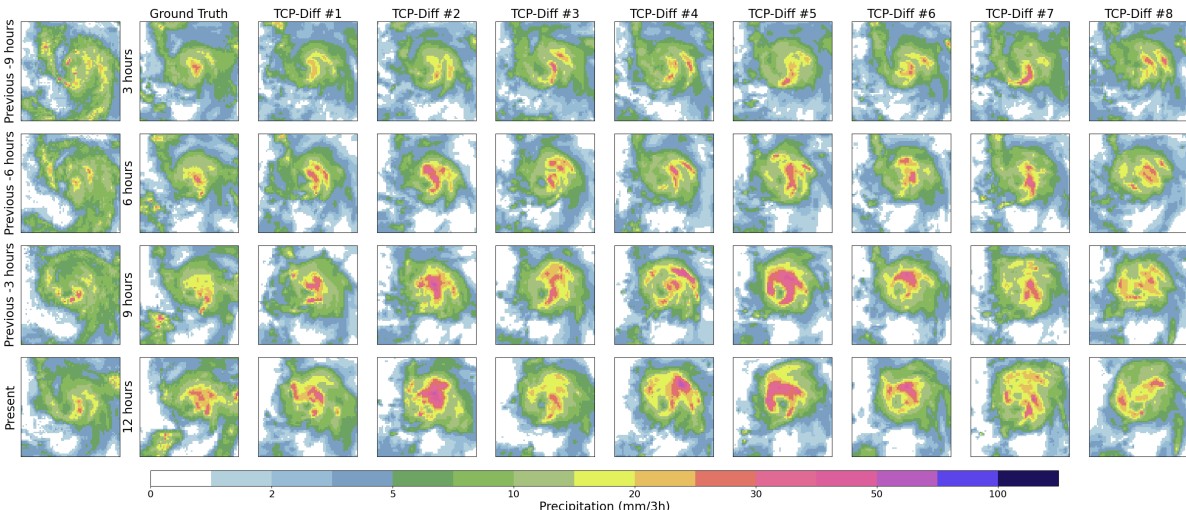

*Figure 9.* The prediction results of different TCP-Diffusion samples on the TC Dumazile in the North Indian Ocean at 03/03/2018 06:00. The first column is the previous 4 timesteps of rainfall data used as input. The second column is the future TC rainfall we want to predict. TCR-Diff#1 to TCR-Diff#8 represent the results of the 1st to 8th tests on this sample.

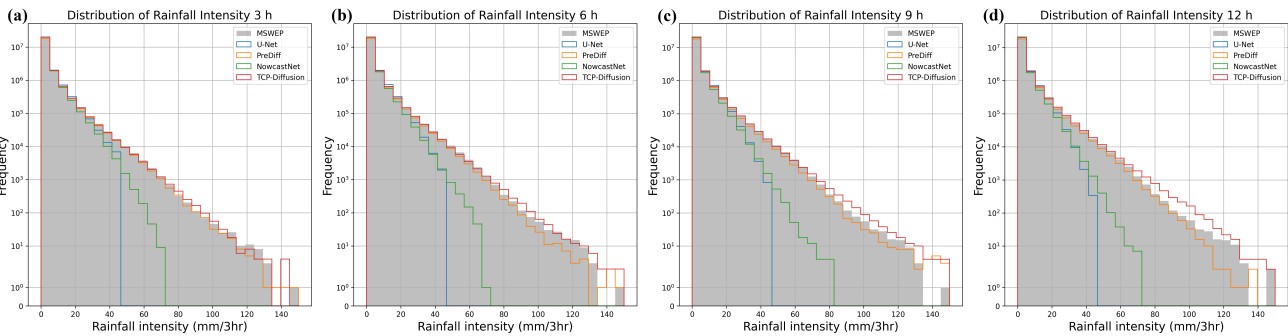

*Figure 10.* TC rainfall forecasting distributions of different DL methods for lead times of 3h (a), 6h (b), 9h (c), and 12h (d). The grey histogram is the distribution of MSWEP observations. The coloured lines show histograms of rainfall rates from different DL forecasting methods. Note the logarithmic vertical axis.

### D.4. Analysis of TC Rainfall Frequency Distribution at Different Lead-times

We also present the TC Rainfall Frequency Distribution predictions for lead times of 3-12 hours, as shown in Figure 10. The results are consistent with those displayed in Figure 4 of the main text, showing that the distribution of predictions by TCP-Diffusion is overall closer to the ground truth rainfall, particularly in the distribution of medium-to-low rainfall intensities, which account for a larger proportion.

### D.5. Analysis of TC Rainfall's Radially Averaged Power Spectral Density at Different Lead-times

We use the Radially Averaged Power Spectral Density to help demonstrate that our model better captures the spatial structure of large-scale TC rainfall systems, as shown in Figure 11. If the power spectral density curve predicted by the model is closer to the ground truth MSWEP at a given spatial frequency, it indicates that the model captures the rainfall variability at that spatial scale more effectively. The low wavenumber region corresponds to large-scale weather systems, such as fronts and tropical cyclones. The RAPSD results of different lead times are similar to that shown in Figure 5.

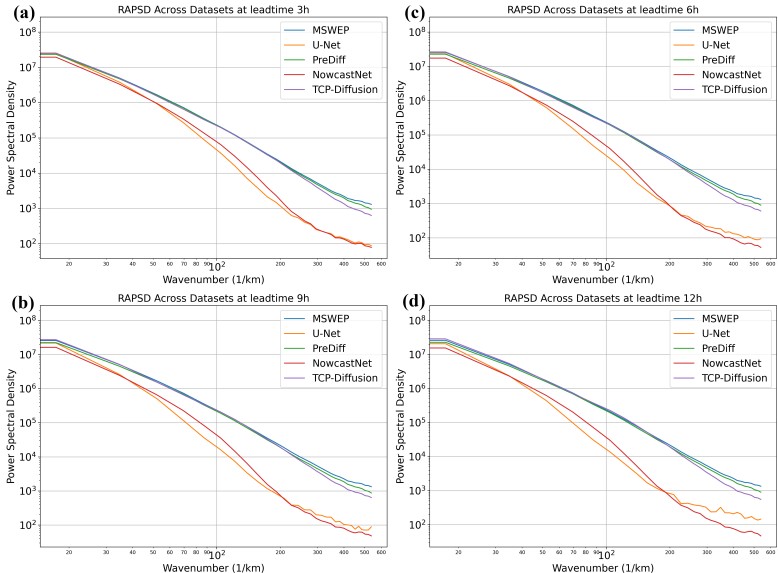

*Figure 11.* Radially Averaged Power Spectral Density (RAPSD) analysis across datasets for lead times of 3h (a), 6h (b), 9h (c), and 12h (d). The plots compare the power spectral density of predicted rainfall using different models over a range of wavenumber. In the low wavenumber region, higher power spectral density represents that the displayed data belongs to large-scale weather systems, such as fronts and tropical cyclones. The power spectral density of TCP-Diffusion in the low wavenumber region is closer to the ground truth rainfall (MSWEP), indicating that our method achieves better capture of rainfall characteristics at the TC scale. Note the logarithmic vertical axis.

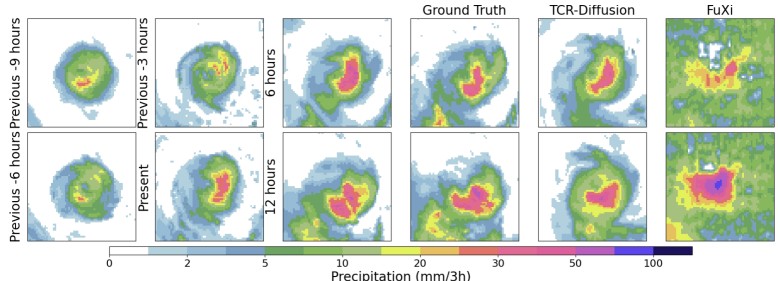

*Figure 12.* The prediction results of TCP-Diffusion and FuXi.

## D.6. Comparison with Large Weather Model

We also compare our model with a state-of-the-art large weather model FuXi(Chen et al., 2023). We obtain the results (6-12h) of FuXi and TCP-Diffusion of 349 samples of the year 2020. Due to differences in the comparison samples as well as the predicted lead times, the TCP-Diffusion data presented in this table vary slightly from those in Table 1 of the main text. As shown in Table 6 and Figure 12, we observe that FuXi performs poorly and tends to significantly overestimate rainfall intensity. We suspect is because FuXi is designed for global variable prediction and thus lacks the ability to capture localized rainfall details associated with tropical cyclones. This highlights the importance of designing models that are specifically centered on tropical cyclone. Such targeted modeling enables a more accurate and localized understanding of rainfall associated with typhoons, which is critical for disaster prevention and early warning.

| Model Name | ETS-6 ↑ | ETS-24 ↑ | ETS-60 ↑ | TP$_{MAE}$↓ |
|---|---|---|---|---|
| FuXi | 0.02770 | 0.03113 | 0 | 5.10396 |
| TCP-Diffusion | **0.39493** | **0.10348** | **0.00489** | **0.46302** |

*Table 6.* The performance of TCP-Diffusion and FuXi on TC precipitation prediction. The forecast diagnostics are the same as in table 1.

| Stages | ETS-6 ↑ | ETS-24 ↑ | ETS-60 ↑ | TP$_{MAE}$↓ |
|---|---|---|---|---|
| Formation | 0.34965 | 0.07112 | 0.00077 | 0.41756 |
| Others | 0.45679 | 0.16220 | 0.00745 | 0.42621 |
| Dissipation | 0.28706 | 0.04400 | 0.00110 | 0.38850 |
| All | 0.43788 | 0.14703 | 0.00644 | 0.42344 |

*Table 7.* The performance of TCP-Diffusion on formation, dissipation, and other phases. The forecast diagnostics are the same as in table 1.

| Intensity | ETS-6 ↑ | ETS-24 ↑ | ETS-60 ↑ | TP$_{MAE}$↓ |
|---|---|---|---|---|
| Tropical Storm | 0.33051 | 0.06312 | 0.00086 | 0.40756 |
| Severe Tropical Storm | 0.39824 | 0.09811 | 0.00286 | 0.40902 |
| Typhoon | 0.43464 | 0.12800 | 0.00695 | 0.42457 |
| Strong Typhoon | 0.49420 | 0.20603 | 0.01046 | 0.46508 |
| Super Typhoon | 0.52014 | 0.23882 | 0.01120 | 0.42191 |
| All | 0.43788 | 0.14703 | 0.00644 | 0.42344 |

*Table 8.* The performance of TCP-Diffusion on different cyclone intensities. The forecast diagnostics are the same as in table 1.

## D.7. Performance of TCP-Diffusion on Different TC Lifecycle-phases

From the results shown in Table 7, we observe that the model performs relatively worse in terms of ETS during the Formation and Dissipation phases. Interestingly, the metric is better in these phases compared to others. This may be attributed to the unstable and irregular rainfall cloud structures during the formation and dissipation stages, which make spatial pattern prediction more difficult. However, rainfall intensity tends to fluctuate less drastically in these stages compared to the active and more intense phases of the TC lifecycle, where extreme values dominate and may introduce higher prediction error. These findings offer useful insights for future work, such as phase-aware modeling or adaptive loss functions.

## D.8. Performance of TCP-Diffusion on Different TC Intensities

From the results shown in Table 8, we observe that the ETS scores consistently improve with increasing cyclone intensity. In contrast, the metric tends to degrade as intensity increases. This pattern is consistent with our earlier observations in the lifecycle-phase analysis 7. A likely explanation is that lower-intensity cyclones (e.g., Tropical Storm and Severe Tropical Storm) tend to exhibit unstable and irregular cloud structures, making spatial rainfall prediction more difficult. However, their overall rainfall intensity is more stable, which may lead to fewer false alarms. On the other hand, higher-intensity cyclones (e.g., Strong Typhoon and Super Typhoon) are structurally more organized, resulting in more predictable rainfall spatial patterns. Yet, the rainfall intensity becomes more variable and extreme, which increases the difficulty of accurate intensity estimation and may negatively affect the metric. These results reinforce the need for adaptive modeling approaches that account for both cyclone intensity and lifecycle phase, which we plan to explore in future work.

## D.9. The Visualizations of Predictions from an NWP Method and TCP-Diffusion

We have shown the comparison of skill scores of ECMWF-IFS and TCP-Diffusion. Here, we also visualize the results from ECMWF-IFS and TCP-Diffusion. As shown in Figure 13, the forecasting results of our method also contain many details. Sample 1 and Sample 2 are different, but their trends of rainfall intensity changes are also similar.

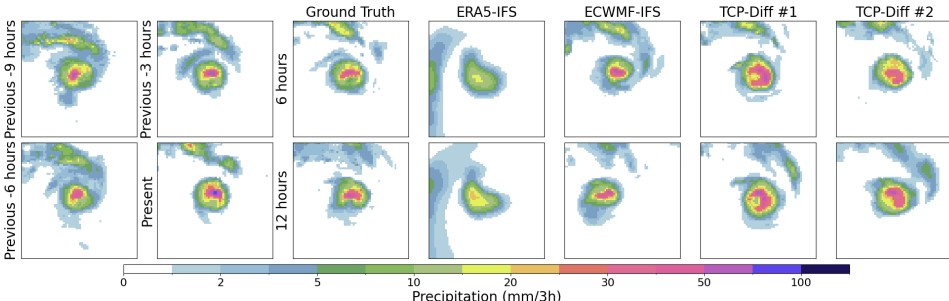

*Figure 13.* The prediction results of NWP (ECMWF-IFS and ERA5-IFS) and TCP-Diffusion on TC Olivia in Eastern Pacific at 03/09/2018 12:00. The first two columns are the past 4 timesteps of rainfall data we input to TCP-Diffusion. The third column is the target observed future TC rainfall. The right columns are the predictions from ERA5-IFS, ECMWF-IFS, and TCP-Diffusion (TCP-Diff #1 and TCP-Diff #2) respectively.

