# OpenReview forum: "TCP-Diffusion: A Multi-modal Diffusion Model for Global Tropical Cyclone Precipitation Forecasting with Change Awareness"
_ICML.cc/2025/Conference — ICML 2025 poster_

### Official Review · Reviewer_5hd2 · 2025-02-26

**Overall Recommendation:** 2

**Summary:**

This paper introduces TCP-Diffusion, a multi-modal diffusion model for tropical cyclone precipitation forecasting. It leverages an Adjacent Residual Prediction (ARP) mechanism to predict rainfall changes, integrates numerical weather prediction data, and employs an Environmentally-Aware 3D U-Net within a diffusion framework. The study evaluates the model against deep learning and numerical baselines, reporting improvements in predicting medium and heavy rainfall.

## update after rebuttal
Thank you for the author's response. However,  from my perspective, the quality of this paper still needs to be improved to meet ICML's standards. Therefore, I will maintain my score.

**Claims And Evidence:**

The paper’s claims are generally supported by experimental results. However, the motivation for using a diffusion model instead of other representation learning-based methods for the prediction task is not clearly justified.

**Essential References Not Discussed:**

The paper appears to have included most of the essential references needed to understand its contributions.

**Experimental Designs Or Analyses:**

The experimental setup follows standard practices, utilizing benchmark meteorological datasets (ERA5, ECMWF-IFS, MSWEP) and established evaluation metrics (ETS, TP-MAE). However, while ARP’s contribution is evaluated, the paper does not analyze how individual encoders (historical vs. future data) affect performance separately. Additionally, while different lead times are tested, a deeper analysis of how performance degrades over time is missing.

**Methods And Evaluation Criteria:**

The paper’s methodological approach is generally well-structured. However, the study presents various comparative experiments, but it does not explicitly assess how well the model adapts to different TC lifecycle stages (e.g., formation, intensification, dissipation).

**Other Comments Or Suggestions:**

N/A

**Other Strengths And Weaknesses:**

**Other Strengths:**

1.	The paper applying diffusion models to tropical cyclone precipitation forecasting is an interesting attempt.
2.	The ARP mechanism helps mitigate cumulative errors and ensures temporal consistency in forecasts.
3.	The topic of applying AI tools for climate research is a hot topic.

**Other Weaknesses:**

1.	The paper describes the use of physical processes only from the results of numerical weather prediction (NWP), but incorporating more TC-related information as model input is an intuitive approach rather than one inspired by physical principles. Instead of seeing a combination of existing deep learning modules for precipitation forecasting, I would prefer to see a novel deep learning design explicitly guided by meteorological physics.
2.	Sensitivity to different cyclone intensities or unseen meteorological conditions is not analyzed, which could impact generalization.
3.	A strong motivation for using a diffusion model over other deep learning approaches (e.g., VAEs, Transformers) for precipitation forecasting is not clearly provided.
4.	I appreciate the authors' effort, but the work lacks sufficient appeal from a machine learning contribution perspective (e.g., introducing new meteorological or physics-informed machine learning approaches). As a result, it may be more suitable for a good meteorology-focused journal.

**Questions For Authors:**

1.	Is incorporating NWP outputs in the model truly reasonable? Traditional numerical weather prediction (NWP) methods involve significant computational costs. If a new deep learning-based approach still depends on NWP results, it does not improve forecasting speed, making it difficult to apply in rapidly changing precipitation prediction scenarios. Could you clarify how this approach balances efficiency and accuracy?
2.	The motivation for choosing a diffusion model over alternatives like VAEs, normalizing flows, or Transformers is not clearly explained. What specific properties of diffusion models make them particularly suited for TC precipitation forecasting?
3.	Have ablation studies been conducted to evaluate the individual contributions of different encoders (historical vs. future data)? How does model performance change when certain encoders are removed or altered?

**Relation To Broader Scientific Literature:**

The paper builds upon existing work by referencing deep learning (DL) applications in meteorology, numerical weather prediction (NWP), and generative models. It builds upon prior work in precipitation forecasting, particularly U-Net-based methods and generative adversarial networks (GANs), and extends recent applications of diffusion models in atmospheric prediction.

**Theoretical Claims:**

No Theorems are presented in this paper.

---

> ### Author Rebuttal · Authors · 2025-03-31
>
> Thank you for acknowledging the contributions of our ARP mechanism. We also understand your concerns regarding the motivation behind using both NWP data and the diffusion model. Below, we provide detailed responses to these questions, and we hope they will help alleviate some of your concerns about our method.
>
> **Q1: A strong motivation for using a diffusion model over other deep learning approaches (e.g., VAEs, Transformers) for precipitation forecasting is not clearly provided.**
>
> We would like to clarify the motivation for choosing diffusion models over other representation learning-based approaches.
> 1. TC precipitation exhibits chaotic behavior and inherent uncertainty, which makes probabilistic forecasting more suitable than deterministic approaches. This is supported by the theoretical discussion in Atmospheric Modeling, Data Assimilation and Predictability, particularly Section 6.5, which states:"*The chaotic behavior of the atmosphere requires the replacement of single ‘deterministic’ forecasts by ‘ensembles’ of forecasts.*" Deterministic models like VAEs can be seen as single forecasts, whereas diffusion models naturally resemble ensemble forecasting by sampling multiple possible futures. This makes them well-aligned with the demands of real-world atmospheric prediction, especially under extreme conditions such as TCs.
>
> 2. Previous methods—including deterministic models like U-Net, VAEs, and Transformer-based architectures—often produce over-smoothed forecasts with limited fine-grained spatial details. We briefly discuss these issues in Lines 69–86 (left column) of the paper. In contrast, diffusion models naturally incorporate noise and denoising processes, allowing them to generate more detailed and realistic outputs.
>
> **Q2: The paper describes the use of physical processes only from the results of NWP, but incorporating more TC-related information as model input is an intuitive approach rather than one inspired by physical principles. (For Weaknesses 1 and 4)**
>
> It is true that our approach adopts a relatively simple yet effective strategy to integrate deep learning with NWP forecasts. However, to the best of our knowledge, this is the first work that combines deep learning with NWP data specifically for TC precipitation forecasting. While the idea may appear intuitive in hindsight, however, the novelty must be evaluated before the idea existed. The inventive novelty was to have the idea in the first place. If it is easy to explain and obvious in hindsight, this in no way diminishes the creativity (and novelty) of the idea.
>
> In our approach, we do not directly embed physical equations into the model (such as PINNs), but instead let the model learn the embedded physical knowledge present in the IFS forecast. This is a simple yet practical alternative to explicitly encoding physical constraints, especially considering the complexity of modeling TC precipitation—a highly nonlinear and chaotic process.
>
> That said, we fully agree that incorporating meteorological physics in a more explicit way (e.g., through constraint-aware loss functions or hybrid physics-ML models) is a promising direction. Bridging the gap between data-driven modeling and physical interpretability is part of our long-term research vision.
>
> **Q3: Is incorporating NWP outputs in the model truly reasonable?**
>
> Reviewer Ndz5 raised a similar concern, which we responded to in Q1. For a more detailed explanation, please refer to that response.
>
>
> **Q4: The ablation study of individual encoders (historical vs. future data) is missing. Additionally, while different lead times are tested, a deeper analysis of how performance degrades over time is missing.**
>
> We would like to clarify that ablation experiments on the historical and future data encoders were indeed included in Table 3. Specifically, the third row in Table 3 represents our model with the ARP and multimodal encoder (M), but without the historical encoders and future data encoders. The fourth row shows the full version of our model, which includes both historical and future encoders. The performance improvement between these two rows demonstrates the effectiveness of integrating NWP forecasts and our encoder designs.
>
> Regarding performance degradation over time, it is a well-known challenge in sequential forecasting tasks. In our setting, this issue is particularly pronounced due to the chaotic nature of atmospheric systems, where small disturbances can amplify over time and lead to substantial divergence in predictions (as described by the butterfly effect). We will provide a more comprehensive analysis in Section D.1 (Line 755) of the camera-ready version.
>
> **Q5: The performances of our method on different TC lifecycle stages and intensities.**
>
> The results are shown at https://limewire.com/d/HHv6Z#HXP582VLSv. Please refer to this link for detailed results.
>
> ***If you have more questions, We'd like to discuss them with you during the author-reviewer discussion period.***

---

### Official Review · Reviewer_Ndz5 · 2025-03-13

**Overall Recommendation:** 2

**Summary:**

This paper addresses two key challenges in medium-range tropical cyclone forecasting: current methods suffer from cumulative errors and the lack of physical consistency. A multi-modal model is proposed, equipped with ARP mechanism to focus on rainfall change to reduce cumulative errors. The integration of NWP system helps to enhance physical consistency.

**Claims And Evidence:**

The flexibility of the method is limited to some extent. Since the future data from an NWP system is necessary, when such data is not accessible, the system seems to stop working. Also NWP system generally requires large-scale computations. If this is the case, it significantly increase the latency for the predictions.

**Essential References Not Discussed:**

The references is good to the reviewer.

**Experimental Designs Or Analyses:**

- More experiments can be conducted on diverse datasets or similar extreme weather prediction such as lightning.
- It is necessary to visualize the residual prediction and show the contribution of how it can reduce cumulative errors.

**Methods And Evaluation Criteria:**

- The overall contribution could be limited. The idea of the adjacent residual prediction is very similar to DiffCast (CVPR2024). What’s your unique model design contributing to TC prediction compared to this method? The idea of using multmodal meteorological factors has been explored in other works such as Pangu and Fengwu.
- Line 258-260, the encoding mechanism is a combination of concatenation and convolution operations. What’s the phylosophy for such computations? Will transformers/attention mechanisms be better? There are many multimodal features fusion methods are available, justifications for the design are expected to be clearly outlined.

Yu, Demin, et al. "Diffcast: A unified framework via residual diffusion for precipitation nowcasting." Proceedings of the IEEE/CVF Conference on Computer Vision and Pattern Recognition. 2024.

**Other Comments Or Suggestions:**

N/A

**Other Strengths And Weaknesses:**

Strengths
- It considers the influence of TC-related meteorological factors and the useful information from NWP model forecasts. Even though DL methods outperform traditional NWP, the results of NWP can still be useful.

Weaknesses
- The significance of the task and input selection is not clear to me, as mentioned in previous review questions. Since the future data from an NWP system is required, when the future data is not accessible, the system may fail to work.

**Questions For Authors:**

Please see my comments in other sections.

**Relation To Broader Scientific Literature:**

This work proposes an idea of using residual information to better capture the spatial and temporal information to medium-range TC precipitation forecasting.

**Theoretical Claims:**

There is no theoretical proofs relevant to this study.

---

> ### Author Rebuttal · Authors · 2025-03-31
>
> Thank you for recognizing the value of integrating our method with NWP. We also understand your concerns regarding the potential impact of using NWP data on the flexibility of our approach. These insights will serve as valuable guidance for our future research. Below, we provide responses to your questions, and we hope they will help address some of your concerns about our method.
>
> **Q1: The flexibility of the method is limited to some extent due to the use of NWP data.**
>
> We appreciate the reviewer’s concern regarding the flexibility and real-world applicability of our method.
> 1. Please note that our model also provides a version that does not rely on IFS forecasts. As shown in Table 3 (Row 3), our model without IFS still outperforms other deep learning baselines listed in Table 1. This is also discussed in Lines 413-418 (right column) of the manuscript. Therefore, our approach remains functional and competitive even in the absence of IFS inputs.
> 2. ECMWF has announced that it will make its forecast data fully open starting in October 2025 (https://www.ecmwf.int/en/about/media-centre/news/2025/ecmwf-achieve-fully-open-data-status-2025). This will significantly reduce the difficulty of accessing IFS forecasts, making it increasingly feasible to incorporate them in practice. We believe our work offers a timely exploration of how to leverage such forecasts effectively in deep learning frameworks.
> 3. we emphasize that IFS and deep learning are not mutually exclusive; rather, their integration can lead to more accurate and efficient forecasts. To our knowledge, this work is the first to integrate NWP forecasts with deep learning for TC precipitation prediction. While IFS captures physical laws through numerical simulation, deep learning excels at learning hidden patterns from data. Our design—which directly feeds IFS outputs into a dedicated encoder for representation learning—offers a simple yet effective fusion strategy. This approach may serve as a useful reference for future efforts in combining NWP with data-driven models across various forecasting tasks.
>
> **Q2: The difference with DiffCast, Pangu and Fengwu.**
>
> We would like to clarify the unique contributions of our model and how they differ from prior works:
> 1. **Difference from DiffCast**: While both our approach and DiffCast involve residual prediction, the underlying concepts are fundamentally different. In our model, we view precipitation forecasting as an accumulative process of rainfall changes over time, where the model directly learns to generate the adjacent residual (future rainfall change) rather than the absolute future value. This is reflected in lines 102-108 (left column).
>
>     In contrast, DiffCast models the residual between a deterministic forecast and the ground truth, as shown in Section 4.2 of their paper. Their diffusion module acts more as a correction mechanism to add details to the output of a deterministic backbone, rather than directly modeling rainfall evolution through temporal differences. Although DiffCast is an inspiring work, our method tackles a different formulation and learning objective. We will discuss this paper in the camera-ready version.
> 2. **Difference from Pangu and Fengwu**: The motivation for using multimodal meteorological inputs in our work is distinct. The prediction targets of Pangu and Fengwu are the future states of multiple variables themselves, hence multimodal inputs are a natural part of the task.
>
>     In contrast, we focus solely on predicting TC rainfall, and we incorporate multimodal inputs to compensate for the limitations of using rainfall data alone, especially in capturing TC rainfall dynamics. Our design emphasizes the integration of environmental and physical information to enhance prediction quality, addressing a gap in existing regular precipitation forecasting models.
>
> 3. **Empirical Validation**:
>     As shown in our ablation results (Table 3), both the proposed ARP and the multimodal input design (M) contribute significantly to model performance. This supports the value of our innovations, particularly in the context of TC precipitation forecasting.
>
> **Q3: It is necessary to visualize the residual prediction and show the contribution of how it can reduce cumulative errors.**
>
> The residual prediction visualizations are shown at this link: https://limewire.com/d/HHv6Z#HXP582VLSv. Please refer to this link for detailed results.
>
> Regarding its role in reducing cumulative errors, predicting residuals rather than absolute rainfall values allows the model to estimate smaller relative shifts at each time step. As a result, even if there are minor prediction errors at individual steps, their cumulative impact is less severe. This approach helps mitigate long-term drift and leads to more stable multi-step forecasts.
>
> ***Owing to space constraints, we would be glad to elaborate further on Methods and Evaluation Criteria (Point 2) or other questions during the author-reviewer discussion period.***

---

### Official Review · Reviewer_caD7 · 2025-03-14

**Overall Recommendation:** 2

**Summary:**

This paper proposes a diffusion model to do precipitation nowcasting relative to a predefined tropical cyclone. The idea is to track the location of a tropical cyclone and to do nowcasting relative to the tracked location. The proposed model also incorporates additional information for the forecasting including IFS, tropical cyclone information, and historical reanalysis. The model acheives a state-of-the-art result compared to other baselines on the newly task.

## Update after rebuttal
I appreciate the efforts of the authors to clarify and address my concerns. Unfortunately, I would not recommend to accept the manuscript in its current version. The manuscript lacks an appropriate comparison to standard baselines for global tropical cyclone precipitation forecasting i.e., using global/regional weather forecast. This comparison should have been made before the main submission and the rebuttal period. The results from the FuXi model are irrelevant since the model was trained on different target.
Moreover, as mentioned in the rebuttal, I would not consider ARP as a novel contribution since many works in weather forecast proposed this technique before. Finally, the  persistence baseline still achieves a similar skill to the proposed model.

**Claims And Evidence:**

- L58-60: Using $\Delta^{t}_{x}$ is a well-known technique for weather forecast see i.e., GenCast (https://doi.org/10.1038/s41586-024-08252-9). However, how Adjacent Residual Prediction (ARP) is going to reduce accumulative errors? $\Delta$ is actually accumulate errors with rollout. That is why some works use continuous or direct forecast i.e., https://arxiv.org/pdf/2312.03876. There is also no ablation study on this claim.

- Without IFS, the prediction is not better than persistence baseline. Compare the first row in Table 1 with the third row in Table 3. I think the improvement comes from the IFS forecast itself since this forecast includes precipitation and correlated variables as well.

**Essential References Not Discussed:**

The paper cited main references reasonably.

**Experimental Designs Or Analyses:**

- I think an ablation about Adjacent Residual Prediction (ARP) or  $\Delta^{t}_{x}$ is currently missing in the paper.

- PreDiff performs worse than a persistence baseline while in the original paper it achieves much higher performance than a persistence baseline (see Table 1 https://arxiv.org/pdf/2307.10422). I think the baselines should be evaluated with out centering or at least they should have a positional encoding to adapt for the task.

**Methods And Evaluation Criteria:**

- Concern about the practicality of the approach for real-world scenario:
As far as I know ERA5 can't be obtained in real-time. IFS forecasts also need time to be generated. I thin the experimental setting is not realistic.

- I think a better evaluation would be to compare to precipitation nowcasting centering around the center of the TC for the proposed model with other baselines that do forecasting but without centering on TC (i.e., this can be done using global forecast).

- Table-2 ECMWF-Ifs: I think the reason why the proposed model is better than IFS is because IFS forecasts the precipitation similar to ERA5. While the proposed model was trained and evaluated against different rainfall Data (MSWEP). It is also not clear if the evaluation was done using total precipitation or just rain.

**Other Comments Or Suggestions:**

Please check equation 6. What type of loss function was used? Do you mean MSE?

**Other Strengths And Weaknesses:**

Strength:
- It is novel to perform precipitation nowcasting relative to tracked trobicla cyclone.
- The paper includes many experiments and ablation studies which make understanding the method clearer.
- The concept of the proposed method is well explained, and the paper is written concisely.

Weakness:
- The model relies a lot on the IFS forecast from ECMWF.
- The paper argues that predicting relative change in precipitation is better than predicting precipitation nowcasting on a global or regional scales. However, there is no experiment to support this argument.

**Questions For Authors:**

1- Why CNN3d was chosen to handle 2D data and a transformer to handle 1D data? And why Resnet? It is not clear from the text i.e., one can use transformer also for 2D data.

2- Do other baselines use the same input information as the proposed model? And what about the IFS forecast, do other baselines use this information?

3- Did you report the baseline scores for the eight test sets similar to the proposed model? see Table 5.

4- Why U-Net is better for ETS-6? Is there any explanation?

**Relation To Broader Scientific Literature:**

Precipitation nowcasting is performed in an absolute sense i.e., over a specific domain or on a global grid. The main contribution of this paper is to do nowcasting of precipitation in relation to a tropical cyclone movement. The novelty is to track the location of the tropical cyclone and then to do prediction relative to the tracked location.

**Theoretical Claims:**

The paper doesn't include proofs.

---

> ### Author Rebuttal · Authors · 2025-03-31
>
> Thank you for recognizing our work, including the novelty of the task itself, the comprehensiveness of our experiments, and the clarity of the manuscript. We also understand the reviewer’s concerns regarding our use of IFS data. Below are our responses to some of the issues raised, and we hope they can help alleviate some of your concerns.
>
> **Q1: The model relies a lot on the IFS forecast from ECMWF.**
>
> 1. Comparing the first row in Table 1 with the third row in Table 3 (the model version without IFS), we can observe that our method still slightly outperforms the persistence baseline overall. Furthermore, even without IFS inputs, our model achieves better performance than other deep learning-based baselines, as also mentioned in Lines 413-418 (right column) of the manuscript.
>
> 2. Our work is, to our knowledge, the first to integrate deep learning methods with NWP (specifically IFS data) for TC precipitation forecasting. As one of the contributions of this work, integrating with NWP can improve the performance of our model, which supports the effectiveness of this contribution and offers useful insights for broader weather forecasting tasks.
>
> 3. The acquisition and usage of IFS data will be further simplified. Please refer to the details in the response to the Q1 of Reviewer Ndz5, point 2.
>
> **Q2: The ablation study of ARP is currently missing.**
>
> It is possible that our explanation of Table 3 was not sufficiently clear. In fact, the ablation study of the ARP mechanism is already presented in the first and second rows of Table 3. The first row corresponds to the original baseline of our method without ARP, while the second row shows the results after incorporating the ARP mechanism. The performance improvement demonstrates the effectiveness of ARP.
>
> **Q3: Using ARP is a well-known technique for weather forecast see i.e., GenCast**
>
> We appreciate the reviewer for bringing GenCast (published in December 2024) to our attention. We have carefully studied the paper and will cite and briefly discuss it in our camera-ready version. Notably, we found that the Residual Prediction in GenCast shares conceptual similarities with our ARP module, which further supports the value of residual modeling in weather-related prediction tasks.
>
> Our ARP design was inspired by the denoising process in diffusion models. Just as diffusion models iteratively refine predictions by removing noise, we view precipitation forecasting as a residual step-by-step accumulation process, as discussed in Lines 102-108 (left column) of our paper. We later found theoretical support for this idea in Chapter 6 of Atmospheric Modeling, Data Assimilation and Predictability, which further reinforced the motivation for adopting ARP in our framework. To the best of our knowledge, this is the first work to apply the ARP mechanism to the task of TC precipitation forecasting. We believe our work can provide inspiration for future research on specialized weather forecasting tasks.
>
> **Q4: Compare with global forecast model FuXi.(For the 2nd point of Methods And Evaluation Criteria)**
>
> We appreciate the reviewer’s suggestion and have supplemented additional experiments to address this point. Among existing large global forecasting models, FuXi (FuXi: a cascade machine learning forecasting system for 15-day global weather forecast) is currently the only one capable of providing global-scale precipitation predictions. However, FuXi does not publicly release official predictions for the total precipitation (TP) variable. So, we reproduced the FuXi model using the official codebase and evaluated its performance on TC precipitation. From the results, we observe that FuXi performs poorly and tends to significantly overestimate rainfall intensity. More details are shown at the link:https://limewire.com/d/HHv6Z#HXP582VLSv
>
>
> **Q5: Concern about the practicality of the approach for real-world scenario. (For the 1st point of Methods And Evaluation Criteria)**
>
> Regarding ERA5, although it is a reanalysis product, near real-time access is possible through collaboration with ECMWF. For instance, Pangu-Weather also utilizes ERA5 reanalysis data and has already been adopted within ECMWF's operational forecasting system for real-time prediction. Therefore, we believe the use of ERA5 in research and applied forecasting scenarios is reasonable and increasingly practical.
>
> For IFS data, the generation time is shorter than that of ERA5 data. Moreover, since we use low-resolution IFS data, the required time is further reduced. This has been mentioned in Lines 427-428 (left column).
>
> **Q6: How ARP is going to reduce accumulative errors?**
>
> A similar concern was raised by Reviewer Ndz5, and we have addressed it in our response to their Q3. Please refer to that response for detailed clarification.
>
> ***Owing to space constraints, we would be glad to elaborate further on Experimental Designs Or Analyses (Point 2) or other questions during the author-reviewer discussion period.***

---

> > ### Comment · Reviewer_caD7 · 2025-04-03
> >
> > > Comparing the first row in Table 1 with the third row in Table 3 (the model version without IFS), we can observe that our method still slightly outperforms the persistence baseline overall. Furthermore, even without IFS inputs, our model achieves better performance than other deep learning-based baselines, as also mentioned in Lines 413-418 (right column) of the manuscript.
> >
> > The model isn't better than the persistence baselines:
> >
> > | Model Name | ETS-6 ↑ | ETS-24 ↑ | ETS-60 ↑ | TP MAE ↓ |
> > | ----------------- | :---------: | :-----------: | :-----------: | :------------: |
> > |Persistence|0.41640|**0.14530** | 0.00564|**0.44558**|
> > |TCP-Diffusion|**0.42926**|0.14253 | **0.00589**|0.44632|
> >
> > In addition, it looks like the baselines are not optimized for the task i.e., a simple U-Net can outperform both PreDiff and NowcastNet models.
> >
> > >  The ablation study of ARP is currently missing
> >
> > Sorry what I mean here is the second point of the weaknesses: The paper argues that predicting relative change in precipitation while centering on TC is better than predicting precipitation nowcasting on a global or regional scales without centering on TC. However, there is no experiment to support this argument.
> >
> > > Novelty of ARP.
> >
> > As mentioned in the review, I would not consider ARP as a novel contribution since SwinVRNN (appeared 2022 and published 2023), GenCast (first appeared in 2023), Graph-EFM (NeurIPS24) and Stormer (first appeared in 2023 and then published at NeurIPS24 https://arxiv.org/abs/2312.03876v1) already used such a technique.
> >
> > > For the 2nd point of Methods And Evaluation Criteria.
> >
> > FuXi model and many other open-sourced global weather forecast models were trained on ERA5 data (which has biases in precipitation), while the target data in this paper is different. The weather forecast baselines should have been trained on the same target data.

---

> > > ### Author Response · Authors · 2025-04-04
> > >
> > > We sincerely appreciate the reviewer’s willingness to engage in further discussion during the author-reviewer discussion period. Below are our detailed responses to the comments raised.
> > >
> > > **Q1: The model isn't better than the persistence baselines. In addition, it looks like the baselines are not optimized for the task i.e., a simple U-Net performers better.**
> > >
> > > Because, the performance gains of our method on ETS-6 and ETS-60 are greater than the performance gaps on the other two metrics, we believe the overall performance is marginally better. It’s worth noting that the persistence baseline achieves relatively strong performance in this specific task, as discussed In Section D.3 of the appendix (“Analysis of Persistence’s Good Performance”)
> > >
> > > Regarding the baselines, they were originally designed for regular rainfall nowcasting. In our re-implementation, we preserved their original parameter settings as much as possible without specific tuning for TC rainfall prediction. Thus, their suboptimal performance in this task highlights a critical issue: methods designed for generic rainfall tasks may not transfer well to the TC rainfall prediction setting. This further underscores the value of developing models specifically for TC rainfall forecasting.
> > >
> > > As for U-Net, it does not consistently outperform other models. In fact, it performs well in light rainfall prediction but performs poorly in heavy rainfall prediction. Prior studies have shown that U-Net can outperform more advanced models under light rainfall. For instance, in PreDiff (https://arxiv.org/pdf/2307.10422), Table 14 (BIAS-16). Moreover, in the paper *Skilful precipitation nowcasting using deep generative models of radar*, U-Net achieves a better CSI-2 (light rainfall) than the proposed DGMR in Fig. 1b. In Fig. 2a, U-Net also performs well for light rain. These results suggest that classical models like U-Net can still be competitive or even superior under certain conditions.
> > >
> > > We also investigated why U-Net achieved a good performance in ETS-6. U-Net’s tendency to generate averaged predictions helps minimize its MSE loss. However, light rainfall dominates in TC rainfall, covering approximately 86.3% of the area around the cyclone center (10°*10°). This causes U-Net to focus more on light rain accuracy, but at the expense of moderate and heavy rainfall prediction, explaining why it fails under heavy rain scenarios.
> > >
> > > **Q2: Second point of the weaknesses and the performance of Fuxi.**
> > >
> > > We would like to clarify that our paper does **not** claim that "predicting relative change in precipitation while centering on TC is better than predicting precipitation nowcasting on global or regional scales without centering on TC," or make any similar assertions. That may have led to our initial misunderstanding of your concern.
> > >
> > > Besides, we have now included a direct comparison with FuXi, a global weather forecast model, during the rebuttal phase. Our method outperforms FuXi not only in quantitative metrics but also in qualitative visualizations (additional visualizations are provided at https://limewire.com/d/HHv6Z#HXP582VLSv).
> > >
> > > We acknowledge that FuXi was trained on ERA5 data, which is known to have biases in precipitation. In contrast, our evaluation is based on the MSWEP V2 dataset (https://www.gloh2o.org/mswep/), which provides more accurate precipitation measurements. Therefore, our model’s better alignment with MSWEP V2 indicates that TCP-Diffusion achieves superior performance in TC rainfall prediction compared to FuXi. This also highlights the importance of developing a TC rainfall-specific model.
> > >
> > > We believe that re-training FuXi or similar large-scale models on MSWEP V2 is neither feasible nor a reasonable requirement. FuXi is capable of producing precipitation forecasts for typhoon scenarios, and directly comparing its outputs with those of TCP-Diffusion is both fair and meaningful. Moreover, re-training FuXi would require substantial computational resources, and FuXi have not publicly released their training code, making such re-training practically infeasible.
> > >
> > > **Q3: Novelty of ARP**.
> > >
> > > We sincerely thank the reviewer for pointing out these recent works and for providing a valuable list of global weather forecasting methods that incorporate similar ARP-like strategies. We acknowledge that we overlooked these references due to the different problem formulations, and we will properly cite and discuss them in the camera-ready version.
> > >
> > > While the use of ARP may seem obvious in hindsight, especially from a general global forecasting perspective, but to the best of our knowledge, **our work is the first to apply an ARP mechanism in TC precipitation forecasting**.
> > >
> > > **We believe that novelty must be evaluated before the idea (using ARP in TC precipitation forecasting) existed. The inventive novelty was to have the idea in the first place. If it is easy to explain and obvious in hindsight, this in no way diminishes the creativity (and novelty) of the idea.**

---

### Official Review · Reviewer_NZjq · 2025-03-14

**Overall Recommendation:** 4

**Summary:**

The article proposes a multimodal diffusion model that integrates data on rainfall, environment, tropical cyclone attributes, and meteorological predictions to generate precipitation due to tropical cyclones globally. Its results outperform other deep learning methods and numerical weather prediction (NWP) models. Among its contributions, the algorithm can predict precipitation globally, demonstrates that forecasting temporal changes in precipitation is more effective, and integrates multidimensional meteorological information.

**Claims And Evidence:**

The authors provide a detailed, clear, and extensive description of the prediction of precipitation changes, the diffusion model, and its solution architecture. The description of their claims is convincing. However, the submission did not include code or data. Having access to them could increase confidence in their results. The authors suggest that the code and data will be available once the article is accepted.

**Essential References Not Discussed:**

Recently, on December 4, 2024, the article "Probabilistic Weather Forecasting with Machine Learning" was published. I believe the results presented in that article would complement some of the elements described in this article.

**Experimental Designs Or Analyses:**

I verified the results and the quantitative and qualitative analysis, the precipitation frequency distribution, and the power spectral density. I did not find any issues with them.

**Methods And Evaluation Criteria:**

The total precipitation mean absolute error, the radially averaged power spectral density, quantifying the precipitation's accuracy, spatial structure, and realism. The paper's metrics make sense in the context of the problem.

**Other Comments Or Suggestions:**

*** I recommend using elements such as \text{Rainfall}_{\text{Current}} when writing text within math environments in LaTeX.

*** ablation for \Delta Rainfall as oppose to total rainfall
*** ablation to computing rainfall everywhere need as opposed to the TC moving center

*** X_t^h is called input data in line 138 (first column). I believe you want to call it historical
** embedding in Figure 2.

*** check equation in line 203, left column

*** use same fonts to name in the text (line 206, first column) and equation (5). Now that you are correcting that, add ',' or '.' at the end of the equations, as needed.

*** change "the following pseudocode" to "the pseudocode in Algorithm 1 (or 2, as needed)"?

*** include input/outputs and variable description in the algorithms

*** pathes, line 259 right column

*** Predif, line 312 right column

**Other Strengths And Weaknesses:**

Overall, I observe a well-formulated article with a strong experimental foundation. Elements such as ARP have been successfully leveraged. The diffusion model for precipitation prediction follows the trend of utilizing them to determine meteorological variables.
I would have liked to review the code and associated data. While the results appear promising, having access to them would have strengthened my confidence in the approach. Additionally, I include some stylistic recommendations later in my review.

**Questions For Authors:**

1. Would it be possible to provide access to at least a subset of the dataset or a
 simplified version of the model during the review process?


2.  Have the authors considered benchmarking their model against probabilistic forecasting
 techniques, especially in terms of uncertainty quantification?




3. How adaptable is TCP-Diffusion to future
improvements in NWP methodologies? Would retraining be necessary with every update to the NWP model,
 or can the framework accommodate updated inputs dynamically?


4. Could the authors clarify whether the model also improves fine-scale precipitation patterns,
 or does it tend to smooth out small-scale variability?

5. Could the authors provide insights into whether there is a feasible way to optimize the
 model's computational efficiency while maintaining its accuracy?

6. Does the model show any systematic underprediction or overprediction of
extreme precipitation events?

**Relation To Broader Scientific Literature:**

The article reviews the related literature, including current approaches based on numerical methods and Deep Learning. I find the review to be comprehensive.

**Theoretical Claims:**

I reviewed the concept of ARP, which is straightforward. The diffusion model is standard and generally accepted. The algorithms on which I am providing comments.

---

> ### Author Rebuttal · Authors · 2025-03-31
>
> Thank you for recognizing our work and for your valuable comments and stylistic recommendations. These suggestions will help us further improve the quality of this manuscript. We will incorporate the corresponding revisions in the camera-ready version. Below are our responses to the issues you raised.
>
> **Q1:Would it be possible to provide access to at least a subset of the dataset or a simplified version of the model during the review process?**
>
> A1:We have created an anonymous GitHub repository (https://anonymous.4open.science/r/TCP-Diffusion-ICML-review/README.md) that includes the testing code and a subset of the test data for our method.
>
> **Q2:Have the authors considered benchmarking their model against probabilistic forecasting techniques, especially in terms of uncertainty quantification?**
>
> A2:We have compared our method with PreDiff, which is also a probabilistic forecasting approach. While we have not explicitly benchmarked uncertainty quantification metrics against PreDiff at this stage, we have evaluated the stability of our predictions. As shown in Table 5, the standard deviation of our method's results is notably small, indicating that our model performs consistently across the dataset.
>
> **Q3:How adaptable is TCP-Diffusion to future improvements in NWP methodologies? Would retraining be necessary with every update to the NWP model, or can the framework accommodate updated inputs dynamically?**
>
> A3:Currently, our model does not dynamically adapt to different NWP methodologies. This is primarily because different NWP models may produce outputs with varying resolutions, spatial-temporal scales, and embedded physical assumptions. Existing offline deep learning frameworks, including ours, typically require consistent input characteristics and cannot yet generalize across heterogeneous NWP outputs without retraining. We greatly appreciate this insightful suggestion—it highlights a direction for making our work more practically applicable. In future research, we plan to incorporate adaptability to diverse NWP models into our framework design.
>
> **Q4:Could the authors clarify whether the model also improves fine-scale precipitation patterns, or does it tend to smooth out small-scale variability?**
>
> A4:Compared to deterministic forecasting methods, our model provides richer details in the prediction of fine-scale precipitation patterns. As illustrated in Figures 3 and 9, especially around the TC center, our method is capable of capturing the shape and structure of heavy rainfall regions. In contrast, deterministic baselines such as U-Net tend to smooth out these localized high-intensity features. This demonstrates the advantage of our probabilistic diffusion-based approach in preserving small-scale variability.
>
> **Q5:Could the authors provide insights into whether there is a feasible way to optimize the model's computational efficiency while maintaining its accuracy?**
>
> A5:As shown in Table 4, our method requires longer training time compared to some traditional deep learning models, and the inference time is also slightly higher than that of non-diffusion approaches. However, given the complexity of the TC precipitation forecasting task, we believe the computational cost of our model remains reasonable. Furthermore, our method is significantly more computationally efficient than NWP systems.
>
> In terms of model design, we adopt a hybrid architecture combining CNNs and Transformers. For high-dimensional data, we utilize CNNs, which are computationally efficient and lightweight. For lower-dimensional inputs, we use Transformers, which, though more computationally intensive, offer stronger feature extraction capabilities—allowing for a more fine-grained understanding.
>
> Additionally, we are exploring model transfer strategies, where a pre-trained large model is adapted to downstream tasks like TC precipitation forecasting. This would allow us to fine-tune only a small portion of the parameters while leveraging the representational power of the large model. This direction represents a promising way to further enhance computational efficiency and is a focus of our future research.
>
> **Q6: Does the model show any systematic underprediction or overprediction of extreme precipitation events?**
>
> A6: We conducted a focused evaluation on extreme precipitation events, which account for approximately 6.73% of the total test samples. The results show that our model tends to slightly overpredict these events, with an average overestimation of about 0.206 mm per 3 hours.
>
> These findings provide valuable insights for future improvements. In particular, we plan to explore techniques such as weighted loss functions or targeted fine-tuning to better handle rare but high-impact precipitation extremes. Addressing this challenge is critical for enhancing the model’s reliability under severe weather conditions.
>
> ***If you have more questions, We'd like to discuss them with you during the author-reviewer discussion period.***

---

### Decision · Program_Chairs · 2025-05-01

**Decision:**

Accept (poster)

**Comment:**

This paper proposes a method to predict tropical cyclone precipitation with a diffusion model. There is substantial disagreement between the reviewers. Reviewer NZjq argues that the paper offers a meaningul contribution and that the paper has sufficient quality. The reviewer engaged in a discussion with the authors, which appears to have clarified the main questions. The other reviewers recommend Weak reject. Reviewer caD7 argues that the comparison with baselines is not strong because the baseline should have been trained on the same data. The other main concern by this reviewer is that Adjacent Residual Prediction (ARP) is not a novel contribution. Reviewer 5hd2 argues that the contribution is limited and that the motivation to use a diffusion model instead of other models is not strong enough. In light of the good results and the fact that the problem tackled has been understudied, I am not too concern about the novelty issues. Even if the methodological novelty is limited, which I will not discuss, that would not justify the rejection of a well-written paper that offers a method with good results on an important application. What is important is that the authors adjust their claims of novelty to reflect the actual literature, as suggested by the reviewers. I would make a similar point about the baselines. The main contribution of the paper, in my view, is to apply machine learning on a important though understudied problem and obtain good results that have not been shown before. Whether similar results could be obtained with other methods is an interesting question but to me not a strong reason for rejection. Regarding the comments about the quality of the paper, my impression is that the paper is well written and other reviewers have not raised this point.

For these reasons, I lean towards recommending accepting the paper despite the average score under the threshold.